

# New Insights From The Jülich Ozone-Sonde Intercomparison Experiments: Calibration Functions Traceable To One Ozone Reference Instrument

Herman G.J. Smit[1], Deniz Poyraz[2], Roeland Van Malderen[2], Anne M. Thompson[3,4], David W. Tarasick[5], Ryan M. Stauffer[3], Bryan J. Johnson[6], Debra E. Kollonige[3,7]

[1]Forschungszentrum Jülich, Institute of Energy and Climate Research, IEK-8: Troposphere, Jülich, 52425, Germany

[2]Royal Meteorological Institute of Belgium & Solar-Terrestrial Centre of Excellence, Uccle, Belgium

[3]Atmospheric Chemistry and Dynamics Laboratory, NASA/GSFC, Greenbelt, MD, USA

[4]University of Maryland Baltimore County, Baltimore, MD, USA,

[5]Environment and Climate Change Canada, Downsview, ON, Canada

[6]Global Monitoring Laboratory, NOAA Earth System Research Laboratory, Boulder, CO, USA

[7]Science Systems and Applications, Inc, Lanham, MD, USA

*Correspondence to*: Herman G.J. Smit (h.smit@fz-juelich.de)



**Abstract**
Although in principle the ECC (Electrochemical Concentration Cell) ozonesonde is an absolute measuring device, in
practice it has several "artefacts" which change over the course of a flight. Most of the artefacts have been corrected in the
recommendations of the Assessment of Standard Operating Procedures for Ozone Sondes Report (GAW Report No. 268),
giving an overall uncertainty of 5-10% throughout the profile. However, the conversion of sampled ozone into the measured
cell current has not been fully quantified, resulting in time-varying background current and pump efficiencies. We describe
an updated methodology for ECC sonde data processing that is based on JOSIE 2009/2010 and JOSIE 2017-SHADOZ test
chamber data. The stoichiometry ($O_3/I_2$) factors and their uncertainties along with the fast and the slow reaction pathways for
the different sensing solution types used in the global ozonesonde network are determined. Experimental evidence is given
for treating the background current of the ECC-sensor as the superposition of a constant ozone independent component ($I_{B0}$,
measured before ozone exposure in the sonde preparation protocol) and a slow time-variant ozone-dependent current
determined from the initial measured ozone current using a first-order numerical convolution. The fast sensor current is
refined using the time response determined in sonde preparation with a first order deconvolution scheme. Practical
procedures for initializing the numerical deconvolution and convolution schemes to determine the slow and fast ECC
currents are given. Calibration functions for specific ozonesondes and sensing solution type combinations were determined
by comparing JOSIE 2009/2010 and JOSIE-2017-SHADOZ profiles with the JOSIE ozone reference UV-photometer
(OPM). With fast and slow currents resolved and the new calibration functions, a full uncertainty budget is obtained. The
time responses correction methodology makes every ozonesonde record traceable to one standard, i.e. the OPM of JOSIE,
enabling the goal of a 5% relative uncertainty to be met throughout the global ozone network.

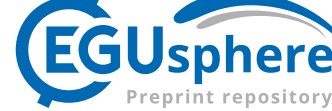

## 1 Introduction

Although it is a minor trace gas constituent of the Earth's atmosphere, ozone plays several essential roles in its chemistry and physics. In the stratosphere, where about 90% of the total ozone amount resides, ozone protects life on Earth by absorbing the harmful ultraviolet (UV) radiation from the sun, adding heat to the stratosphere. In the upper troposphere, ozone is an important absorber of infrared radiation, acting as a powerful greenhouse gas (IPCC-Climate Change, 2013, 2023). Ozone is the primary source of the hydroxyl (OH) radical in the troposphere, controlling the lifetime of hundreds of pollutants (Seinfeld and Pandis, 2016), and determining its oxidizing capacity (Thompson, 1992). The stratosphere is a natural source of tropospheric ozone but approximately half of the ozone in the troposphere is formed photochemically when combustion (vehicular, industrial or pyrogenic) processes release $NO_x$, $(NO + NO_2 = NO_x)$, carbon monoxide (CO) and hydrocarbons (also referred to as volatile organic compounds (VOC) that react through free radical cycles in the presence of UV. VOC may also originate from combustion or natural sources, the latter predominantly from vegetation and to a lesser extent from the ocean. Surface ozone is considered a pollutant with adverse impacts on human and animal health (e.g., respiratory problems) and on vegetation (Mills et al., 2018) and is a primary marker for "Air Quality," setting the scale for Good, Fair, and Unhealthy definitions used by local Air Quality agencies (Garner and Thompson, 2013). The photochemistry of ozone pollution or "smog" was worked out in the 1950s (Seinfeld and Pandis, 2016); surface ozone measurements became widespread as regions or nations enacted regulations to mitigate episodes of high ozone.

Measurements of stratospheric ozone gained attention in the 1960s and 1970s when it was recognized that natural levels of ozone were regulated by catalytic cycles involving nitrogen oxides ($NO_x$, $N_2O_5$, $NO_3$ and $HNO_3$), hydrogen oxides (with $H_2O$ vapor a source of OH and $HO_2$, $HO_x = OH+HO_2$) and halogens (XO and $XO_2$, where X was Cl or Br derived from oceanic methyl chloride and methyl bromide). Anthropogenic perturbations of these cycles were investigated when it was recognized that emissions of N- and Cl-containing compounds by rockets and high-altitude aircraft could threaten stratospheric ozone (Crutzen, 1970; Stolarski and Cicerone, 1974). A worse threat was hypothesized when it was realized that chlorofluorocarbons (CFCs) present in the atmosphere (Lovelock et al., 1973), but relatively inert in the troposphere could enter the stratosphere and destroy ozone photochemically there (Molina and Rowland, 1974). Perturbed stratospheric ozone chemistry by CFCs was a cause for alarm, leading to first regulations in CFC usage in the 1970s. However, it was not until ground-based total ozone monitoring (Farman et al., 1985) discovered catastrophic springtime ozone loss over Antarctica in 1984-1985 that international action was taken to phase out Ozone Depleting Substances through the 1987 signing of the Montreal Protocol (UNEP-Ozone Secretariat,14[th] edition, 2020). Implementation of the Montreal Protocol and its follow-on Amendments require governments to monitor ozone, reporting every four years to the World Meteorological Organization (WMO) and United Nations Environment Programme (UNEP) in Scientific Assessments on total column ozone, its vertical distribution and attribution of long-term. Since 1991 there have been nine UNEP/WMO Scientific Assessments, with the most recent report released in 2022 (WMO/UNEP, 2023).

Global monitoring of total ozone has relied on satellite instruments since the 1970s but ground-based instrumentation deployed on all continents still provides ground-truth. In particular, ozonesondes are essential for satellite algorithms and validation of satellite-derived profiles and reanalysis products (Wang et al., 2020; Thompson et al., 2022). Balloon-borne ozonesondes, flown together with radiosondes, make relatively inexpensive, accurate, all-weather measurements of the ozone concentrations from the ground to 30 km or higher, with ~100 m vertical resolution (Smit, 2014). The electrochemical concentration cell (ECC) ozonesonde has been deployed for more than 50 years with ~60 stations currently launching on all continents (GAW Report No.268, 2021; Thompson et al., 2022; Stauffer et al., 2022). Ozonesonde data constitute the most important record for deriving ozone trends throughout both stratosphere and troposphere, particularly in the climate-sensitive altitude region near the tropopause where satellite measurements are most uncertain. Strategic ozonesonde networks like MATCH and IONS (Intensive Ozonesonde Network Studies) have been organized to support aircraft campaigns in



characterizing photochemical and dynamical interactions affecting vertical and regional ozone distributions (Thompson et
al., 2007a and 2011; Tarasick et al., 2010).
**1.1 Establishing Quality Assurance/Quality Control (QA/QC) practices for ozonesondes (1996-2021)**
Despite the advantages of ozonesonde profiles, there is a challenge in that each ozonesonde instrument is unique, typically
launched only once, and it must be carefully prepared prior to launch in order to obtain accurate data. Processing of the final
measurement is carried out using certain parameters determined pre-launch. In addition, there are two manufacturers of
ozonesondes that show systematic offsets relative to each other. Further biases in ozonesonde datasets can occur because
three variants of the sensing solution that produce the ECC current signal from the ozone are currently in use. The
ozonesonde community has created guidelines for operations and data processing applicable to the range of instrument and
sensing solution types used in the global ECC-sonde network. When the guidelines are followed it is possible for
consistently high-quality data to be collected across the global network.
The creation of guidelines or "best practices" has evolved over the past 20 years in a process referred to as the Assessment of
Standard Operating Procedures (SOP) for Ozonesondes (ASOPOS) and organized through the WMO Global Atmosphere
Watch (GAW). The key element of ASOPOS was the establishment of the World Calibration Centre for Ozone Sondes
(WCCOS) with a custom-designed Environmental Simulation Facility (ESF) at the Research Centre in Jülich, Germany, in
1995 (GAW Report No.104, 1994; Smit et al., 2000). The ESF consists of an absolute ozone measuring reference, a fast
response (2s), accurate (2-3%), dual beam UV-absorption ozone photometer (OPM) (Proffitt and McLaughlin, 1983)
attached to the chamber that enables control of pressure, temperature and ozone concentration simulating flight conditions of
an ozone sounding up to 35 km over ~ 2 hours (Smit et al., 2007). Up to four ozonesonde instruments at once can be
intercompared through this process. Simulations in the ESF included conditions of polar, midlatitude, subtropical and
tropical sonde launches. Other aspects of sonde operations, e.g., response times to rapid changes in ozone concentration, are
also tested in the ESF. Since 1996, nine Jülich OzoneSonde Intercomparison Experiment (JOSIE) campaigns have been
conducted at WCCOS and documented in a series of publications (Smit and Kley, GAW Report No. 130, 1998) for JOSIE-
1996; JOSIE-1998 (Smit and Sträter, GAW Report No. 157, 2004a), JOSIE-2000 (Smit and Sträter, GAW Report No. 158,
2004b; Smit et al., 2007; Thompson et al., 2007b); JOSIE-2009/2010; JOSIE-2017 (Thompson et al., 2019). The first three
JOSIEs, which tested several non-ECC instruments as well as Science Pump Corporation (SPC) and ENSCI ECC
instruments, showed the ECC-sonde to be more accurate. After JOSIE-2000 only ECC-sondes were tested in the WCCOS.
In 2004 a the WMO/BESOS (Balloon Experiment on Standards for OzoneSondes) field campaign, carried out in Laramie
(Wyoming, USA) deployed a large gondola with 18 ozonesondes and the OPM of WCCOS (Deshler et al., 2008) with results
similar to JOSIE-2000. These early experiments demonstrated that high precision and accuracy depend not only on sonde
manufacturer and sensing solution strength, but also on pre-launch preparation details. Smit et al. (2007) concluded that
standardisation of operating procedures for ECC sondes yields a precision better than ± (3-5) % and an accuracy of about
±(5-10)% up to 30 km altitude.
In 2004 an expert team of ozonesonde operators, data providers and manufacturers formally instituted the ASOPOS to
analyse the results of BESOS and the JOSIE campaigns up to that time. The ASOPOS goal was to ensure consistency of data
quality across stations and within individual station time series by specifying how to prepare and operate the ozonesonde
instrument and to accurately process and report profile data. The first set of SOP recommended by ASOPOS, based on the
JOSIE campaigns from 1996 to 2000 and BESOS, was published online in 2012 and as GAW Report No. 201 in 2014 (Smit
and ASOPOS 1.0 Panel). To make (historical) ozonesonde time series records compliant with the ASOPOS standards, an



OzoneSonde Data Quality Assessment (O3S-DQA) activity was initiated in 2011 within the framework of SI2N[1], resulting in
procedures for "homogenizing" data and estimating uncertainties (Smit and O3S-DQA Panel, 2012; https://www.wccos-
josie.org/o3s-dqa); transfer functions in support of the guidelines were documented in Deshler et al. (2017). Within several
years roughly half of the global network stations had reprocessed their data (Tarasick et al., 2016; Van Malderen et al., 2016;
Thompson et al., 2017; Sterling et al., 2018; Witte et al., 2017, 2018, 2019; Ancellet et al., 2022). Comparisons between
original and homogenized data showed that significant systematic errors were eliminated, particularly where changes in
technique and/or equipment had been made.
The homogenised time series were based on having raw currents from the ozonesonde cells, a prerequisite for the analysis
and processing methods of the present paper. However, the ozonesonde community agreed that several issues were
unresolved. These included the complexity of the so-called "background current" characterized during the preparation and
the lack of traceability of the archived ozone profile to an absolute standard. A JOSIE-2017 campaign was designed to
address these concerns. In addition to the tests of prior JOSIEs, the 2017 tests focused on a single regime, tropical profiles, to
gather a larger set of statistics. A special challenge of tropical soundings is that near the tropopause the ozone signal to noise
is typically very small, giving artefact low readings (Vömel et al. 2020). JOSIE-2017 (also called JOSIE-SHADOZ) was
carried out with eight SHADOZ operators who supplied their home-prepared sensing solutions, following their own
preparation procedures for half the simulations (Thompson et al., 2019). The other half of the simulations tested a lower-
buffer variant of the sensing solution with the WMO/GAW SOP. The overall results of JOSIE-2017 resembled those of the
1996-2000 JOSIE and BESOS. In other words, the offsets of the various instrument-sensing solution types (SST) from the
OPM reference and associated biases of ECC sonde instruments and SST had not changed over more than 20 years.
An ASOPOS 2.0 Panel formed in 2018 to review the JOSIE-2017 campaign data along with lessons learned from
reprocessed datasets and the JOSIE 2009/2010 results. ASOPOS 2.0 published GAW Report No. 268, "Ozonesonde
Measurement Principles and Best Operational Practices" (Smit, Thompson and ASOPOS, 2021; hereafter referred to as
GAW Report No. 268) as an update to GAW Report No. 201. The newer report gives the same recommendations as GAW
Report No. 201 on sonde manufacturer-SST combinations, but stricter and more unified SOP. The latter consist of more
detailed recommendations based on physical principles of the ozonesonde measurement. More explicit procedures are given
for data quality indicators, hardware usage and maintenance and metadata. GAW Report No. 268 also specified for the first
time how to report ozone profiles traceable to the standard OPM. However, the issues of a time-varying background current,
specification of uncertainties in the ozone measurement (and related pump efficiencies) required analysis beyond GAW
Report No. 268 before consensus could be reached on data-processing recommendations. That is the scope of this paper.
**1.2 Addressing residual ozonesonde QA/QC issues from WMO/GAW 268. Outline of paper**
Chapter 3 of GAW Report No. 268 draws on the Tarasick et al. (2021) review of ozonesonde performance characteristics.
Both documents point out that the greatest barriers to reducing uncertainties in the final ozone measurement derive from (1)
the use of improper pump efficiencies and (2) a background current that varies with ozone exposure (hence with time) over
the course of the balloon ascent. The current paper revisits fundamentals of the ozonesonde measurement to overcome these
two shortcomings. The here reported methodology to resolve the fast and slow time responses builds on an earlier study by
Imai et al. (2013), and more recently on the work by Tarasick et al. (2021) and Vömel et al. (2020). We first give a more
detailed description of the physical and chemical origin of the ECC ozonesonde signal (Section 2), illustrated with laboratory
measurements from the Uccle, Belgium, ozonesonde station. Section 3 first corrects for the background signal composed of

---

[1] This is a joint initiative under the auspices of SPARC (Stratosphere–troposphere Processes And their Role in Climate), the International Ozone Commission (IO3C), the ozone focus area of the Integrated Global Atmospheric Chemistry Observations (IGACO-O3) programme, and the Network for Detection of Atmospheric Composition Change (NDACC). For simplicity, an acronym of acronyms, SI2N, was adopted.



(i) a constant physical component ($IB_0$) and (ii) a small and slow varying (time constant 25 min) chemical component that
varies with ozone exposure. The remaining fast component of the signal is then corrected by deconvolution with an
exponential decay with a time constant between 20 and 30s. Although the approach is similar to Vömel et al. (2020), an
advantage of our updated method is that it is developed from and applied to dedicated JOSIE chamber data (JOSIE
2009/2010) that used consistently prepared ozonesondes, with detailed in-flight and post-flight measurements and metadata.
Second, the simultaneous OPM measurements in the simulation chamber serve as reference data for determining key
parameters of the method, e.g. the contribution of the slow component to the overall signal. In Section 4, the OPM reference
data are used to evaluate the updated method with comparisons to the conventional method. For these analyses,
measurements from all JOSIE campaigns, covering a range of simulated environments are used. Comparing residuals of the
corrected ozonesonde profiles to the OPM profiles allows us to determine a set of the calibration functions for each
instrument-SST combination (Section 5) and to estimate uncertainties of the updated time response correction (TRC) method
(Section 6). The TRC method is implemented with actual sounding data in Section 7 for ascent and descent profiles at
tropical, mid-latitude and polar (Antarctic) stations and improvements with respect to the conventional approach are
quantified. A summary and outlook appear in Section 8.
**2 Physical and Chemical Origins of the ECC Ozonesonde Signal**
**2.1 Principle of Operation**
The ECC (=Electrochemical Concentration Cell) ozonesonde, developed by Komhyr (1969), uses an electrochemical method
to measure ozone which is based on the titration of ozone in a neutral buffered potassium iodide (NBKI) sensing solution
according to redox reaction:

$2 KI + O_3 + H_2O \rightarrow I_2 + O_2 + 2 KOH$                (R1)


A neutral pH≈7 is obtained through the addition of a phosphate buffer ($NaH_2PO_4.H_2O$ and $Na_2HPO_4.12H_2O$)
The titration involves a coulometric method employing electrochemical cells to determine the amount of generated "free"
iodine ($I_2$) per unit time through conversion into an electrical current at a depolarizing cathode electrode. The actual ECC
component of the ozone sensor, made of Teflon or molded plastic, consists of two chambers. Each chamber contains a
platinum (Pt) mesh electrode that serves as cathode or anode. The chambers are immersed in a KI-solution of different
concentrations and linked together to provide an ion pathway and to prevent mixing of the cathode and anode concentrations.

Continuous operation is achieved by a small nonreactive gas sampling pump (Komhyr 1967) forcing ozone in ambient air
through the cathode cell that contains a lower-concentration KI-sensing solution, causing an increase of "free iodine" ($I_2$)
according to the redox reaction (1). Transported by the stirring action of the air bubbles, the free $I_2$ contacts the Pt-cathode
and convert to 2 $I^-$ through the uptake of two electrons. At the Pt-anode surface, $I^-$ is converted to $I_2$ through the release of
two electrons. The overall cell reaction is:

$3 I^- + I_2 \rightarrow I_3^- + 2 I^-$                (R2)


The electrical current $I_M$ (µA) generated in the external circuit of the electrochemical cell is directly related to the uptake rate
of ozone in the sensing solution. By knowing the gas volume flow rate $\Phi_{P0}$ [$cm^3s^{-1}$] of the air sampling pump and its
temperature $T_P$ (K), the electrical cell current $I_M$ (µA), after subtracting a background current $I_B$ (µA), is converted to the
ozone partial pressure $P_{O3}$ (in mPa) (Komhyr 1969):




$$P_{O3} = 0.043085 * \frac{T_P}{(\eta_P * \eta_A * \eta_C * \Phi_{P0})} * (I_M - I_B) \qquad (1)$$

The constant 0.043085 is determined by the ratio of the universal gas constant, R, to twice the Faraday constant, F, (because
two electrons flow in the electrical circuit from reaction (R2) (Komhyr 1969).
The overall efficiency of conversion consists of:

a) Pump efficiency, $\eta_P$, that declines at lower pressures. At reduced air pressures (< 100 hPa), the pump efficiency

declines due to pump leakage, dead volume in the piston of the pump, and the back pressure exerted on the pump
by the cathode cell (Komhyr 1967, Steinbrecht et al., 1998, Nakano and Morofuji, 2023).

b) Absorption (i.e capture) efficiency, $\eta_A$, for the transfer of the sampled gaseous ozone into the liquid phase. Although

evaporation reduces the amount of the sensing solution available for ozone uptake, $\eta_A$ is not significantly affected
(Komhyr, 1971). Thus, $\eta_A$ remains at 1.0, with an uncertainty of < ±1% (Tarasick et al., 2021; Davies et al., 2003).

c) Conversion efficiency, $\eta_C$, of the absorbed ozone in the cathode solution creating iodine that leads to the measured

cell current $I_M$. Historically, it has been assumed that $\eta_C$ is unity at neutral pH (Saltzman and Gilbert, 1959;
Komhyr, 1969; Komhyr, 1986). However, there is now a great deal of evidence that this is not quite the case, as will
be discussed below.

Currently, there are two manufacturers of ECC ozonesondes, Science Pump Corporation and Environmental Science
Corporation, most recently producing the SPC-6A and EN-SCI-Z ozonesonde series, respectively. The designs of both ECC
types are similar but differences include: (i) the material of the electrochemical cell (Teflon for SPC-6A and molded plastic
for EN-SCI-Z); (ii) ion bridges (details are not known due to manufacturer proprietary issues); (iii) layout of the metal
frame. Since 2014, a modified ECC-type ozonesonde manufactured at the Institute of Atmospheric Physics (IAP), Beijing,
has been produced (Zhang et al., 2014a,b) but to date, few comparisons of the Chinese instrument with the well-
characterized SPC-6A and EN-SCI models have been carried out. Thus, profiles from Chinese instruments are not included
in the current study.

Three different aqueous sensing solution types (SST) are commonly used in the ECC-sonde cathode cells: (i) SST1.0: 1.0%
KI & full buffer; (ii) SST0.5: 0.5% KI & half buffer; (iii) SST0.1: 1.0% KI & 1/10th buffer (GAW Report No. 268),
respectively. In all cases a KI saturated cathode solution is employed in the anode cell. Laboratory studies by Johnson et al.
(2002) found that, depending on the concentration of the cathode sensing solution, the stoichiometric ratio of the ozone to
iodine conversion reaction (1) can increase from 1.00 up to 1.05-1.20. Johnson et al. (2002) determined that this increase is
caused primarily by the phosphate buffer and to a lesser extent depends on the KI concentration. No significant influence of
KBr-concentration was observed, although its role is not well understood. From JOSIE 2000 (Smit et al., 2007), BESOS
2004 (Deshler et al., 2008) and multiple other sounding tests (e.g. Deshler et al., 2016) it is known that there is a significant
difference in the ozone readings when sondes of the same type are operated with different sensing solutions, e.g. STT0.5 and
SST1.0. Both sonde types exhibit a systematic change of sensitivity, about 5-10% over the entire profile, when the sensing
solution is changed from SST0.5 to SST1.0. Johnson et al. (2002) demonstrated that this offset is mostly caused by the
phosphate buffer with a minor contribution from the KI- concentration. In addition, the EN-SCI sonde tends to measure
about 4-5 % more ozone than the SPC-sonde when operated with the same SST for reasons that are not understood.
**2.2 Impact of Pump efficiency and Conversion Efficiency (Stoichiometry)**
The accuracy of the ECC ozonesonde depends on the extent of the ozone-iodide reaction in the cathode cell and the
efficiency of the reduction of the iodine produced, which can be expressed primarily in the overall uncertainty based on the



contribution of the individual uncertainties of each parameter expressed in Eq. (1). Tarasick et al. (2021) quantified and
reviewed the uncertainty budget of the measured partial pressure of ozone, confirming that the most critical parameters are
the (background) current for the tropospheric part of the ozone profile and the pump and conversion efficiencies used in the
post flight data processing for the stratospheric part of the ozone profile.

Since JOSIE 1996 (Smit and Kley, 1998) it was recognized that, if the preparation and data correction procedures prescribed
by Komhyr (1986) are used, an increase of the stoichiometric factor, presumably due to evaporation of the cathode sensing
solution in the course of the sounding, may be compensated by a too low pump flow correction in the stratosphere above 20-
25 km altitude. With new pump flow calibrations and stoichiometry investigations, Johnson et al. (2002) demonstrated that
the pump efficiency tables reported by Komhyr (1986) and Komhyr et al. (1995) indeed compensate for the increase of the
stoichiometric factor, i.e. the conversion efficiency. Commonly used pump efficiencies and their uncertainties recommended
by ASOPOS 2.0 (GAW Report No. 268) are listed in Table 1.

**Table 1:** Pump efficiencies ($\eta_P$) as a function of air pressure for ECC ozonesondes reported by (i) Komhyr (1986) and referred
to as K86; (ii) Komhyr et al. (1995), called K95; (iii) Johnson et al. (2002), referred as NOAA/CMDL & UWYO at
Univ.Wyoming; (iv) Nakano and Morofuji, 2023, at JMA.

| Pressure [hPa] | ECC (SPC-6a) *Komhyr,*1986 K86-Efficiency | ECC (ENSCI) *Komhyr et al.,* 1995 K95-Efficiency | ECC (CMDL) *Johnson et al.,* 2002 | ECC (UWYO) *Johnson et al.,* 2002 | ECC (JMA) *Nakano and Morofuji,* 2023 |
|---|---|---|---|---|---|
| 1000 | 1 | 1 | 1 | 1 | 1 |
| 100 | 0.989 ± 0.005 | 0.993 ± 0.005 | 0.968 ± 0.009 | 0.978 ± 0.011 | 0.978 ± 0.009 |
| 50 | 0.985 ± 0.006 | 0.982 ± 0.005 | 0.951 ± 0.011 | 0.964 ± 0.012 | 0.964 ± 0.011 |
| 30 | 0.978 ± 0.008 | 0.972 ± 0.008 | 0.935 ± 0.011 | 0.953 ± 0.015 | 0.948 ± 0.013 |
| 20 | 0.969 ± 0.008 | 0.961 ± 0.011 | 0.918 ± 0.012 | 0.938 ± 0.018 | 0.929± 0.014 |
| 10 | 0.948 ± 0.009 | 0.938 ± 0.021 | 0.873 ± 0.015 | 0.893 ± 0.026 | 0.883 ± 0.017 |
| 7 | 0.935 ± 0.010 | 0.920 ± 0.022 | 0.837 ± 0.019 | 0.858 ± 0.029 | 0.848 ± 0.020 |
| 5 | 0.916 ± 0.012 | 0.889 ± 0.021 | 0.794 ± 0.023 | 0.817 ± 0.034 | 0.807 ± 0.023 |


The pump efficiency tables reported by Johnson et al. (2002) and more recently by Nakano and Morofuji (2023) are both
based on a large number of pump calibrations using complementary and well-established methods. Both tables are generally
consistent within statistical uncertainty, but diverge significantly from the older Komhyr (1986) and Komhyr et al. (1995)
tables. Although they have historically been called "pump efficiencies", the Komhyr values in Table 1 are now recognized as
empirical efficiencies, which combine pump efficiency and conversion efficiency for the standard buffered solutions SST1.0
and SST0.5 (Tarasick et al., 2021). For consistency with long-term data records, the values reported by Komhyr (1986) and
Komhyr et al. (1995) are recommended by ASOPOS 2.0 (GAW Report No. 268) for SPC-6A & SST1.0 and EN-SCI &
SST0.5, respectively.

Normally, in the pH = 7 buffered KI sensing cathode the stoichiometry of the conversion (R1) of ozone into iodine is
assumed to be 1.00 with an uncertainty of about ±0.03 (Dietz et al., 1973), while the initial absorption efficiency of gaseous



ozone into the sensing solution will be 1.00 with an uncertainty of 0.01. These values for $\eta_A$ and $\eta_C$ are used in the
conventional method of ozonesonde data processing as recommended by ASOPOS in GAW Report No. 268 and before in
GAW Report No. 201.

**2.3 Perspectives on the Background Current**


**2.3.1 $I_{B0}$ and $I_{B1}$ Conventions for Background Currents**


The ECC sensor background current, $I_B$, is defined as the residual current output by the cell when sampling ozone free air.
Since the 1990s during the preparation of the ECC sensor at the day of flight, two background currents, $I_{B0}$ and $I_{B1}$,
respectively, are measured: before and after exposure of a certain amount of ozone, usually about 5µA ozone equivalent for
about 10 minutes. Both background currents are measured after flushing the cell for 10 minutes with ozone free air. (GAW
Report No. 201 and GAW Report No. 268). Although small (typically < 0.1 µA), the ECC sensor background current may be
of appreciable magnitude compared to the current when there is very low ozone such as in the tropical upper troposphere or
in the stratosphere above 5 hPa but also during ozone hole conditions in polar regions.

Background measurements of SPC-5A sondes operated with the SST 1.0 using ozone-free air, showed before about 1993,
typical values of $I_{B0} = 0.06\pm0.02$ µA and $I_{B1} = 0.09\pm0.02$ µA, respectively (Smit, 2004c). After 1993 $I_{B0}$ dropped to values of
0.00-0.03 µA and consequently $I_{B1}$ dropped by about 0.06 µA. This may mean that the manufacturer made changes, most
likely cleaning or conditioning the electrodes or ion bridge (e.g. less leakage of $I_2$ into the cathode solution). In the past thirty
years, both SPC-6A and EN-SCI sondes show similar low $I_{B0}$ and $I_{B1}$ values when a high-quality gas filter flushes the cells
with ozone free "zero" air. However, the difference of $I_{B1}$-$I_{B0}$ of ~ 0.03-0.04 µA has stayed the same over decades. This is
actually the "chemical" contribution of the overall $O_3$ + KI chemistry in the cathode cell to the measured background current
after zero-air flushing, whereas $I_{B0}$ is independent of ozone exposure and assumed to be an inherent property of the ECC-
sensor. The latter has been demonstrated in several laboratory experiments (Smit et al., 2007; Vömel and Diaz, 2010), and in
this study (Sect.2.3.3).
**2.3.2 Constant Background Current?**
In the early days of the ECC there was no clear distinction between $I_{B0}$ or $I_{B1}$ to apply for $I_B$ in Eq. (1). Komhyr (1969)
suggested that $I_B$ resulted largely from a residual sensitivity of the ECC sensor to oxygen, such that $I_B$ decreased with air
pressure in proportion to the rate at which oxygen entered the sensor. Thornton and Niazy (1982) showed in a laboratory
study that the primary source of the background current is from the removal of residual tri-iodide, normally present in the
cathode solution and not from the reaction of oxygen with iodide to produce tri-iodide nor from the direct reduction of
oxygen. Since 1975 the manufacturer (Science Pump Corporation) has preconditioned the ECC electrodes with iodide such
that the oxygen dependence has become vanishingly small and can be neglected (Thornton and Niazy, 1982).

Theoretically, an ECC sensor in electrochemical equilibrium will produce no current; any current in the absence of ozone or
other oxidants must be due to an imbalance of tri-iodide between the anode and cathode cells (Komhyr, 1969). Possible
causes of such an imbalance include (i) a leaky ion bridge, (ii) limited mass transfer of residual tri-iodide ($I_3^-$) in the cathode
solution (Thornton & Niazy, 1982), (iii) limited electron transfer at the cathode surface, (iv) an imbalance resulting from cell
conditioning or contamination, or (v) previous exposure to ozone. The first three cases represent a background current that
may be expected to remain roughly constant and should therefore be subtracted as a best approximation; however, the last
two cases, (iv) and (v), should decline according to the response time of the cell (Tarasick et al., 2021).



### 2.3.3 Past Ozone Dependent Background Current

Based on simulation chamber experiments Smit et al. (1994) recommended using $I_{B0}$ for the constant $I_B$ subtraction, which was confirmed in a field experiment by Reid et al. (1996). However, the results could not be confirmed in later JOSIE experiments which demonstrated that the background current most likely varies with the past ozone measured, implying that two background currents operate over the sonde operation (Smit and Sträter, 2004a,b; Smit et al., 2007): (i) one background current $I_{B0}$, which is independent of ozone exposure and (ii) a second past ozone dependent background current that will vary in the course of the sounding. This time variant ECC background current is assumed to result from a minor, but still slowly decaying, contribution to the measured cell current. Based on laboratory experiments Johnson et al. (2002) and Vömel and Diaz (2010) suggested that its origin is related with the ECC-chemistry having a fast (20-30 s) and an additional minor pathway (reaction time constant ~20-30 min) that causes a memory effect, probably due to slow side reactions in the oxidation of iodide by $O_3$ in the cathode sensing solution. In equilibrium this can lead to an overall stoichiometry factor, $O_3/I_2$, larger than 1.0. The magnitude of the excess stoichiometry depends strongly on the phosphate buffer concentration in the cathode sensing solution. Vömel and Diaz (2010) suggested that, instead of a measured background current, it would be better to use an appropriate solution dependent conversion efficiency and background current values in the basic ECC-formula Eq. (1). For improved data processing the contributions of the slow (20-30 min) and fast (20-30 s) responses to the overall measured ECC ozone signal need to be considered simultaneously using an appropriate response (memory) function.

Such a possible methodology may be the deconvolution of the measured ozone profile after determining the overall frequency response of the combined sensor and air sampling system (De Muer and Malcorps, 1984). However, the method is complicated and not practical to be apply to the global ozonesonde network. More accessible are first order numerical schemes that deconvolve the fast response which were developed and tested by Imai et al. (2013) and Huang et al. (2015). Tarasick et al. (2021) further developed one simple first order numerical scheme to resolve both the fast and slow time responses of the ECC-sensor. Vömel et al. (2020) developed the methodology for quantifying the fast and slow currents in more detail but several aspects were not fully considered and their methodology was not assessed with the most comprehensive data base and for various pairs of sonde types and SSTs. This study remedies these gaps.

To investigate the chemical origins of the slow current, laboratory response-time tests for hundreds of ECC-ozone sensors (EN-SCI, SST0.5) were made at the Uccle (Belgium) sounding station since August 2017 during every routine day-of-launch preparations to measure the two time constants in the ECC signal. In this experiment, the following steps were taken to record the ECC sensor current as function of time:

    a. Before ozone exposure, flush the ECC-cell for 10 min with zero air: Record $I_{B0}$.

    b. Expose the ECC-cell for 10 min to 5 µA ozone equivalent.

    c. Flush the ECC-cell for 10 min with zero air: Record $I_{B1}$ and stop flushing (pump inactive, short-circuit sensor leads)

    d. No Flushing until t= 55 min, then flush 5 min. zero air: Record $I_{B60}$ and then stop flushing.

    e. No Flushing until t=115 min, then flush 5 min with zero air: Record $I_{B120}$.

The steps (a) to (c) follow exactly GAW Report No. 201 and GAW Report No. 268 SOPs. However, after these steps, most of the time between t=10 and 120 min., flushing with ozone-free air has stopped except for the 5-minute periods at t=55 min and t= 115 min. During the 5 minutes of flushing a short current increase was observed but it declined rapidly with a typical "fast" 1/e response time of 25 seconds. The 120-min timing was chosen because this is the typical duration of the ascent of an ozone sounding. Summaries of the observations for the fast and slow currents appear in Figure 1.



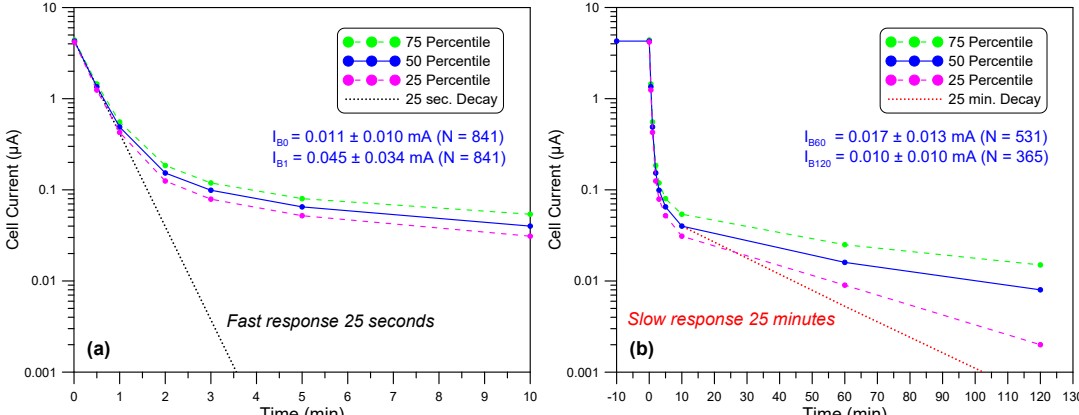

**350**

**351**  **Figure 1.**    Relaxation of the ECC-cell current (logarithmic scale) flushed with purified ozone free air as function of time

**352**  after the cells have been exposed for 10 minutes with 5 µA ozone. The sequence: (i) No flushing t=10-55 min.; (ii) Flushing

**353**  t=55-60 min.; (iii) No flushing t=60-115 min; (iv) Flushing t= 115-120 min. Left diagram first 10 minutes relaxation (grey

**354**  dotted line: 1/e decay of $I_M$(t = 0 min) with 25 s. time constant) and right diagram show the full two hours of relaxation (red

**355**  dotted line: 1/e decay of $I_{B1}$ (t = 10 min.) with 25 min. time constant).

**356**

**357**  The observed relaxations in Figure 1 follow a typical superposition of two first order exponential decays of the fast and the

**358**  slow component which can be expressed here as:

**359**  $$I_M(t) = I_{F0} Exp\left[\frac{-t}{\tau_F}\right] + I_{S0} Exp\left[\frac{-t}{\tau_S}\right] + I_{B0} \qquad (2)$$

**360**  where $I_{F0}$ and $I_{S0}$ are the fast and slow sensor current contributions, respectively, at the start of the response test at t=0.

**361**

**362**  Although, after t=10 min. until t=120 min. only two short periods of 5 minutes the cathode cell was flushed with ozone free

**363**  air, the results are fairly consistent with the observations of Vömel and Diaz (2010), who flushed the cathode cell over the

**364**  entire 120 minutes relaxation period. Clearly the relaxation of the slow component of the background is independent of the

**365**  flushing, i.e. no stirring action in the cathode sensing solution, and therefore most likely has a chemical origin from a slow

**366**  reaction pathway. The $I_{B0}$ and $I_{B1}$ shown in Fig.1 are typical of present-day ECC sondes (e.g. GAW Report No. 268). Further,

**367**  the characteristic difference of $I_{B1}$ and $I_{B0}$ of about 0.03-0.04 µA has been observed over a large number of sondes ($\cong$800)

**368**  and is most likely the residual of the slow reaction pathway.

**369**

**370**  In contrast to Vömel and Diaz (2010), based on around 25 runs, in the more than 350 Uccle experiments the cell current do

**371**  stabilize after 1-2 hours decay time to the background current before exposure to ozone, $I_{B0}$. As a matter of fact, assuming a

**372**  25 min 1/e-decay from the mean $I_{B1}$ = 0.045 µA at t=10 min, the $I_{B60}$ and $I_{B120}$ would decay on average down to 0.006 µA

**373**  and 0.00055 µA, after 60 and 120 minutes, respectively. Actually, we recorded mean values of 0.017 µA and 0.01 µA,

**374**  respectively. The average differences of $I_{B60}$-$I_{B0}$ and $I_{B120}$-$I_{B0}$ are 0.008 µA and < 0.001 µA, respectively. Similar

**375**  observations were made in 1993 in the simulation chamber at WCCOS, whereby four ECC sondes were flushed for more

**376**  than 90 minutes with zero ozone air during the simulation of a tropical descent pressure profile. After a relaxation time of

**377**  about 70 minutes the cell currents approximate constant values which are very close to the corresponding recorded $I_{B0}$ (for

**378**  details see Fig. S1 in the supplementary material). This means that after 1-2 hour of flushing the ECC-sensor with zero

**379**  ozone, the remaining current is identical to $I_{B0}$, so that during the typical duration of the ascent of an ozone sounding, the



remaining current ($I_{B0}$) persists, which is not the result of a 25 min decay but has another origin. This inherent $I_{B0}$ of the
ECC-sensor, possibly caused by a small leakage of iodine ($I_2$) from the ion bridge into the cathode solution or by a mass-
transfer limit in the solution or electron transfer at the cathode surface (Thornton and Niazy, 1982, 1983), is assumed
constant over the 2 hours of an ozonesounding.

To understand the KI+$O_3$ chemistry and the impact of the phosphate buffer on the stoichiometry of the conversion of the
sampled ozone into "free" iodine, Tarasick et al. (2019, 2021) reviewed many studies in which a variety of KI-solution
strengths with different pH-buffers were investigated. The reaction mechanism of KI+$O_3$ in aqueous solution in presence of a
phosphate buffer as investigated by Saltzman and Gilbert (1959) may explain the observations made here and are discussed in
detail in Appendix A. In short, they proposed two reaction pathways: a primary reaction pathway without a buffer and the
secondary pathway with a buffer. Experimentally, Saltzman and Gilbert (1959) showed that the impact of the slow reactions
increases with the buffer concentration, whereas buffered solutions with no KI showed no evidence of any $O_3$ reactions. This
means that the additional reactions with $O_3$ are secondary reactions after the initial $O_3$ + KI reaction. Saltzman and Gilbert
further demonstrated that the secondary pathway can form additional free iodine, half of it reacting very fast (<< than 1 sec,
i.e. residence time of air sample in the cathode cell), the other half more slowly (~25 min). This means that the secondary
reaction pathway can contribute both to the fast and slow ECC current, respectively. However, loss mechanisms may occur
too. In summary, we do not know exactly the stoichiometry of the fast and slow reaction pathways leading to "free" iodine."
Therefore, we can only indirectly quantify these two stoichiometries that lead to the fast and slow cell current components
observed, respectively. In other words, the measured cell current $I_M(t)$ is the superposition of

$$I_M(t) = I_{P,F}(t) + I_{S,F}(t) + I_S(t) + I_{B0} \qquad (3)$$
where
$I_{P,F}$ = sensor current contribution from fast primary reaction pathway.
$I_{S,F}$ = sensor current contribution from fast secondary reaction pathway.
$I_S$ = sensor current contribution from slow secondary reaction pathway with a typical 20-25 min time response.
The contribution of the fast reaction pathways that form iodine fast is lumped together in the total fast sensor current
component $I_F(t)$ with a typical time response of 20-30 s. The measured sensor current $I_M(t)$ is then expressed as:
$$I_M(t) = I_F(t) + I_S(t) + I_{B0} \qquad (4)$$
The overall stoichiometry $S_T$ of the chemical conversion of $O_3$ into $I_2$ is the sum of the stoichiometry factors $S_F$ and $S_S$ of the
fast and slow reaction pathways, respectively.

**2.4 Formulating New Fast and Slow Components of the ECC Current**
From the response tests (fast decay from 5µA down to 0.1-0.5µA within less than 1 minute) it can be concluded that $S_F$ is
close to one (0.9-1.1) and at least a factor 10-20 larger than $S_S$, which is small (0.01-0.10). The time scale of the slow current
component ($\tau_S$=25 min) is about a factor of 60 slower than the dominating fast current component. This means that the slow
current acts as a slowly time-varying background current. The latter can be treated as a superposition with the ozone-
independent background $I_{B0}$ to constitute to the total background, but given now as the time varying $I_B(t)$ in Eq. (1).
$$I_B(t) = I_{B0} + I_S(t) \qquad (5)$$
By substituting $I_M(t)-I_B(t)$ into Eq. (1) the partial pressure of ozone is now expressed as Eq. (6):
$$P_{O3} = 0.043085 * \frac{T_P}{(\eta_P * \eta_A * \eta_C * \Phi_{P0})} * I_F(t) \qquad (6)$$
where the fast sensor current is expressed as:

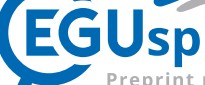

$\qquad I_F(t) = I_M(t) - I_S(t) - I_{B0}$ (7)
The conversion efficiency may depend on sonde type and sensing solution type. It is largely related to the stoichiometry of
the conversion of $O_3$ into $I_2$ from the primary fast reaction pathway and to a lesser degree on the secondary reaction pathway.
The partial ozone pressure can be determined from equation Eqs. (6)-(7) in two steps:
a.   Determine the slow current as function of time. Because the past ozone exposure-dependent slow current

component $I_S(t)$ is much slower and smaller than the fast current component $I_F(t)$, the slow current can be

determined from the convolution of the measured current $I_M(t)$ with the slow time constant $\tau_S$=25 min.

b.   Calculate the fast current $I_F(t)$ and then through deconvolution of $I_F(t)$, resolve the time delay of the relatively fast

time constant $\tau_F$=20-30 seconds.

The fast as well as the slow reaction path are determined by a first order time response and can therefore be separated in a
convolution part to determine $I_S(t)$ and a deconvolution part to obtain the fast current component, $I_{F,D}(t)$, respectively. The
mathematical techniques used here to resolve the impacts of the slow and fast time constants, $\tau_S$ and $\tau_F$, respectively, are
based on the numerical scheme described by Miloshevich et al. (2004), and were first applied by Imai et al (2013) to resolve
the time delay effects caused by the ECC fast response time. A first order response of a measured sensor signal $U$ (here ECC
ozone sensor current) that is approximately exponentially to a change in $U$, is described by the common "growth law
equation":
$\qquad \frac{dU_m}{dt} = \frac{1}{\tau} * (U_a - U_m)$ (8)
where $U_m$ is the instantaneous measured signal, $U_a$ is the ambient ("true") signal that is driving the change in $U_m$, and $\tau$ is the
time constant of the signal.
Integrating Eq.(8) over a small time step $\Delta t_k = t_{k-1} - t_k$ gives the measured signal as a function of time:
$\qquad U_m(t_k) = U_a(t_k) - \{U_a(t_k) - U_m(t_{k-1})\} * Exp\left(-\frac{\Delta t_k}{\tau}\right)$ (9)
This assumes that the time step $\Delta t_k$ is small relative to the response time $\tau$. Further, it is assumed that the "true" (ambient)
signal $U_a$ is quasi-stationary during time step $\Delta t_k$ such that $U_a(t_k) = U_a(t_{k-1})$. The exponential term is the response function.
Eq. (9) can be expressed in a numerical convolution or de-convolution scheme. From Eq. (9) we can obtain $I_S(t)$ and $I_{F,D}(t)$,
as follows:
***Case 1: Slow current component derived from convolution (time constant*** $\tau_S$***) of the ambient sensor current*** $I_a$***:***
To obtain the slow current component ($I_S$), $U_m$ in Eq. (9) is substituted by the slow fraction of $I_a$, represented here by the
stoichiometry $S_S$ multiplied with the ambient ("true") ozone sensor current $I_a$. Eq. (9) can now be re-written into the
integrating form:
$\qquad I_S(t_k) = S_S * I_a(t_k) - \{S_S * I_a(t_k) - I_S(t_k - 1)\} * X_S$ (10)
whereby the slow response function $X_S$ is:
$\qquad X_S = Exp\left(-\frac{\Delta t_k}{\tau_S}\right)$ (11)



***Case 2***: Deconvolution (time constant $\tau_F$) *of the fast signal $I_F$ with* $\tau_F$:
To obtain the deconvolved fast current component $I_{F,D}$ , Eq. (9) should be solved to obtain $U_a$ (=$I_{F,D}$), and $U_m$ is substituted by
the fast fraction $I_F$. Eq. (9) can then be re-written into the differentiating form:
$$I_{F,D}(t_k) = \frac{I_F(t_k) - I_F(t_{k-1})*X_F}{(1-X_F)} \tag{12}$$

where the fast response function $X_F$ is:
$$X_F = Exp\left(-\frac{\Delta t_k}{\tau_F}\right) \tag{13}$$


Compared to Vömel et al. (2020), the recursive numerical convolution scheme proposed here (Eq.11) is the same, while the
deconvolution scheme (Eq.12) differs through the inclusion of the exponential fast response function $X_F$ (Eq. 13) itself,
rather than its first order approximation. The latter allows larger time steps $\Delta t_k$, which may become significant for older
ozone sounding records that had data with resolution of 10 seconds or more.
**3 Resolving Slow- and Fast-Response Signals using JOSIE 2009/2010**
To resolve the slow and fast time responses of the measured ECC sensor current, the JOSIE measurements conducted in
several campaigns between 1996 and 2017 form an ideal dataset, because of several reasons. Firstly, all the ozonesonde
preparations and the measurements were carried out in a controlled environment. Secondly, the availability of simultaneous
reference measurements from a fast-response photometer OPM with high precision and accuracy provide an absolute
reference for the derived ozone profiles. Further, in the course of the simulation several response tests are performed in
which the ozonesondes and the OPM are exposed to zero-ozone air for a five minutes period (see Fig. 2). These response
tests enable us to determine the stoichiometry of the slow reaction pathway and subsequently the slow sensor current $I_S(t)$ as
a function of time. In this sense, the JOSIE 2009 and 2010 campaigns dataset is of particular interest, because all
experiments included four of those response tests in the simulation profiles themselves.

For the sake of clarity, it is to be noted that the here reported ozone readings of the OPM are already based on the new UV-
absorption cross-section, referred to as the CCQM.O3.2019 (BIPM, 2022; Hodges et al., 2019) value that is about 1.23%
lower than the former cross-section (Hearn et al., 1961) that was mostly used before in the global ozone ground based
monitoring networks. Consequently, all $P_{O3}$ measurements of the OPM reported here are about 1.23% larger than the values
reported before in earlier JOSIE-publications.
**3.1 JOSIE 2009/2010**
The JOSIE 2009 and 2010 protocols are similar to the JOSIE 1998 campaign (Smit and Sträter, 2004a; Smit et al., 2007). In
2009 a set of 40 brand new ECC sondes (20 SPC6A and 20 ENSCI) were tested; in 2010 the same set of ECC sondes, re-
furbished and tested under the same conditions, were evaluated against the same OPM reference. One aim of these
campaigns was to test the performance of brand new and refurbished ozonesondes. It was found that the re-used sondes
agree within 1%–2% with brand new sondes, although with a slightly lower precision of ~5% (see Fig. 3.1 in GAW Report
No. 268). The JOSIE 2009/2010 ozonesondes were prepared by only three operators, strictly following the same preparation
protocols, including the use of purified air from the same cylinders for the ozone-free air source. It can therefore be
considered as an ideal data set for well-prepared ozonesondes. All ozonesonde data were processed according to the
guidelines of GAW Report No. 268, which we denote as the "conventional" method hereafter. That means: (i) subtracting the
constant background current $I_{B1}$; (ii) correcting the pump flow rate for the moistening effect; (iii) using the pump flow rate
efficiency correction tables Komhyr (1986) and Komhyr et al. (1995) for SPC and EN-SCI ozonesondes respectively; (iv)





converting the measured pump temperature to the internal pump body temperature, with an additional small pressure
dependent correction (GAW Report No. 268); and (v) no total ozone normalisation. Note also that all simulations were
identical in representing a typical mid-latitude ozone profile (Smit et al., 2007).
During both campaigns, a total of 26 simulation runs were made, of which all but one had 4 ozonesondes simultaneously in
the simulation chamber, giving a total amount of 103 ozonesonde profiles. However, 17 of those profiles were gathered
using research-mode SSTs and are not included here. Fourteen simulations were carried out in December 2009, 2 in January
2010, and 10 in August 2010.
**3.2 Determination of Slow Current $I_S(t)$**
**3.2.1 Determination of Stoichiometry $S_S$**
To determine the relative contribution $S_S$ of the slow component in the ECC ozonesonde signal, in other words, the
stoichiometry factor of the slow reaction pathway of conversion of $O_3$ into $I_2$, the response tests of the JOSIE 2009/2010
dataset are used. Four time response tests are included during these simulations at four different pressure levels, (RT1: 475-
375 hPa, RT2: 100-85hPa, RT3: 20-15 hPa, RT4: 6-5 hPa), during which ozone-free air is provided in the simulation
chamber for 5 minutes. A typical example of a JOSIE 2009 simulation run is given in Figure 2. After 5 minutes the fast
sensor current has declined by more than 16 1/e relaxation times and is vanishing small. This means that at the end of this
time response test, the only contribution to the overall measured current $I_M(t)$, after correction for $I_{B0}$, comes from the
remaining slow current component. At this moment, the fast co-existing OPM data (red in Fig. 2) provides the measure of
the true value of the ozonesonde signal. The next paragraphs outline the different practical steps.
To obtain a direct measure of the true ECC-ozone sensor current, the OPM ozone partial pressure is converted to the generic
OPM current ($I_{OPM}$) for each individual ozonesonde using sonde pump temperature, sonde pump flow rate and pump
efficiency values of JMA (Nakano and Morofuji, 2023, See Table 1), as in Eq. (1).

$$I_{OPM} = \frac{(\eta_P * \eta_A * \eta_C * \Phi_{P0})}{T_P * 0.043085} * P_{O3,OPM} \qquad (14)$$


In other words, we are calculating the generic sensor current corresponding to the ozone equivalent measured by the OPM,
as if it were the true ECC ozone current. This means that the generic $I_{OPM}$ is taken as the actual reference ("true") current for
determining the slow stoichiometry factor $S_S$.

Additionally, the generic OPM current $I_{OPM}$ (red in Fig. 2) is convolved into $I_{OPM,C}$ with an exponential time response with $\tau_s$
= 25 minutes using Eq. 9, to obtain a slow time response into the generic OPM current signal (yellow in Fig. 2).
$$I_{OPM,C}(t_k) = I_{OPM}(t_k) - \{I_{OPM}(t_k) - I_{OPM,C}(t_k - 1)\} * X_S \qquad (15)$$

Finally, the slow stoichiometry factor $S_S$ is obtained by taking the ratio of the remaining ECC sensor current $I_M$ minus the
constant background current $I_{B0}$ and the convolved OPM signal ($I_{OPM,C}$), at the end of the time response test intervals RT1,
RT2, RT3, RT4, when only the slow component is expected to contribute to the sonde signal, such that
$$S_S = \frac{(I_{M(ECC)} - I_{B0})}{I_{OPM,C}} \qquad (16)$$




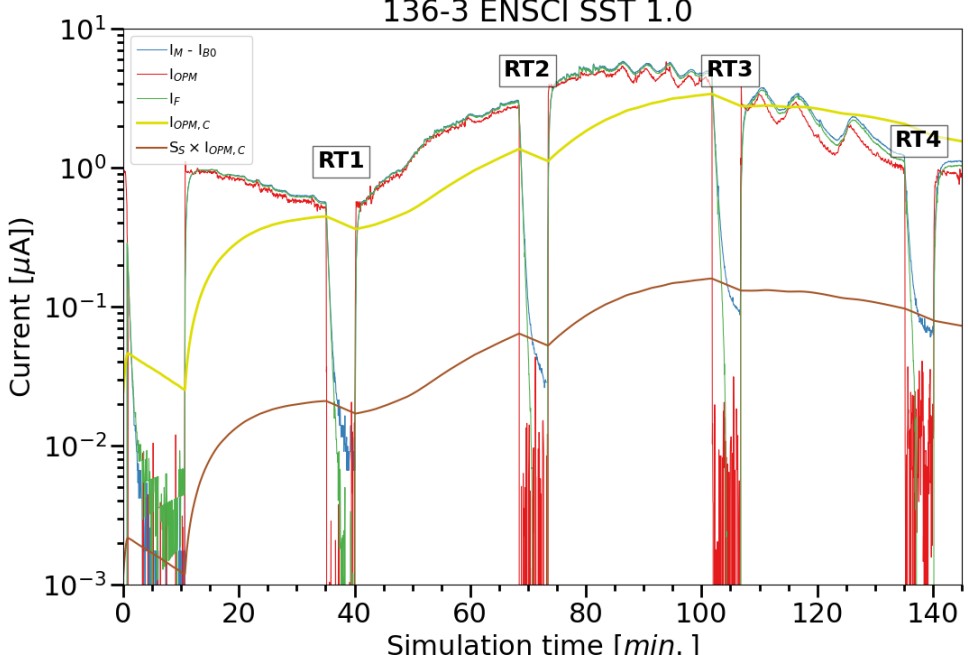



**Figure 2.** Example of a simulation run during JOSIE 2009 as a function of the simulation time, with the measured ECC
current $I_M$ minus $I_{B0}$ (blue line), the generic OPM current $I_{OPM}$ (red line), the 25 min convolved $I_{OPM,C}$ (yellow line) and the
25 min convolved $I_{OPM}$ adapted to $I_M$-$I_{B0}$ after the determination of the slow stoichiometry factor $S_S$ or slow current $I_S$ (= $S_S$ x
$I_{OPM,C}$ ) (brown line) and the fast sensor current $I_F$ (green line), obtained after correction of the measured sensor current $I_M$ for
the constant background current $I_{B0}$ and the slow current contribution $I_S$

The ratios used to obtain the slow stoichiometry factor ($S_S$) values are calculated during the final 50 seconds of each time
response test, RT1, RT2, RT3, RT4, respectively. Those values, obtained for all ozone profiles within each sonde type and
SST combination, are shown in Fig. 3, together with median, 25th and 75th percentile values. The median $S_S$ values and their
Median Absolution Deviation (MAD) uncertainties are given in Table 2. Note that the determination of the mean $S_S$ values
(and their uncertainties) is very robust, and does not depend on the time response test interval or the slow time lag constant.
We will come back to this in Sect. 6.2.  Further it showed that by varying $\tau_S$ = 25 min. by ± 5 min. the corresponding $S_S$
values only changed by less than 5%, which is small compared to the MAD uncertainty of $S_s$ (Table 2).

The most striking feature is that $S_S$ only depends on the SST, not on the sonde type. This confirms our hypothesis on the
origin of this slow component, as described in Section 2.4. For SST0.5 and SST1.0 there is an almost proportional relation
between the magnitude of $S_S$ and the buffer strength, independent of the KI concentration (or percentage). This result has
been explained by the secondary reaction pathway of the reaction mechanism after Saltzman and Gilbert (1959), whereby the
extra slow stoichiometry contribution is caused by the buffer (Appendix A). However, a comparable result does not hold  for
SST0.1 (Table 2). One would expect that for the low buffered case (SST0.1) $S_S$ should be much smaller than for the SST0.5.
This is not true; $S_S$ is even slightly larger. It seems that for the SST0.1, other competing reaction mechanisms may occur,
which do depend on the KI concentration, and may generate free iodine on a 25-minute time scale. Such a hypothetical





mechanism may also explain the fact that for low or no buffered SST we still measure $I_{B1}$ background currents with values of
0.01-0.03 µA larger than $I_{B0}$ as measured in JOSIE 2000 (no buffer SST; Smit and Sträter, 2004b) and JOSIE 2017 (SST0.1;
Thompson et al., 2019). A speculative mechanism is that the electronically excited oxygen singlet molecule formed in (R3)
of the primary reaction pathway of the $O_3$+KI chemistry (Appendix A) may, in addition to de-activation in (R4), react with
$H_2O$ and produce hydrogen peroxide ($H_2O_2$) (e.g. Xu et al., 2002). The formed $H_2O_2$ would oxidize KI to produce free
iodine, but on a time scale of 25 minutes which could contribute to the slow current $I_S(t)$. Further studies are required to
understand the underlying chemical processes.

**Table 2**: Median and their Median Absolute Deviation (MAD) uncertainty values of the slow stoichiometry factor $S_S$
obtained from JOSIE 2009 and 2010 for SPC and EN-SCI ozonesondes operated with the sensing solution types SST0.5 and
SST1.0. The stoichiometry factor $S_S$ for EN-SCI/SST0.1 has been determined with the same approach but using laboratory
measurements at Uccle with an ozone reference instrument (see Appendix B).  *: the same value for SPC/SST0.1 has been
adopted as for EN-SCI 1.0%-0.1B. $N_S$ is the number of sonde profiles.

| Sonde Type | SST1.0 | SST0.5 | SST0.1 |
|---|---|---|---|
| SPC | $0.050 \pm 0.002$ ($N_S$ =16) | $0.017 \pm 0.004$ ($N_S$ =21) | $0.023 \pm 0.005$* |
| EN-SCI | $0.046 \pm 0.006$ ($N_S$ =23) | $0.018 \pm 0.004$ ($N_S$ =15) | $0.023 \pm 0.005$ ($N_S$ =8) |


The stoichiometry factors $S_S$ (Table 2) to determine the slow current $I_S(t)$ are substantially lower than the so called "steady
state bias factors" applied by Vömel et al. (2020). These steady state bias factors were determined as the overall excess
stoichiometry to one from laboratory experiments with a fixed ozone exposure during several hours (Figs. 3 & 4 in Vömel
and Diaz, 2010). In this study we derived for SST1.0 $S_S = 0.046$-$0.050$ which is only half the 0.09 value of Vömel et al.
(2020). For SST0.5 and SST0.1, our respective $S_S = 0.017$-$0.018$ and 0.023 values are also smaller than their 0.024 and 0.031
steady-state bias factors. Using the same laboratory procedures as Vömel et al. (2010), Johnson et al. (2002) reported an
excess overall stoichiometry of ~0.07 for SST1.0. The lower factors obtained in this study, particularly for SST1.0, might
also be related to the different methodology followed for determining $S_S$. Here, $S_S$ values are determined from the response
of a downward step under zero-ozone conditions. In Johnson et al. (2002), and Vömel and Diaz (2010) the excess
stoichiometry factors were determined from the relatively small differences observed between the ECC sonde and a
reference UV-photometer after a 60-min upward step ozone exposure. The latter requires very accurate generation of ozone
values with a precision better than 1% to determine the relatively small excess stoichiometry factors involved. Note also that
for the earlier studies reference ozone readings are based on older UV absorption cross sections that are now corrected by
1.23% to be compatible with the new UV absorption cross-section applied to the OPM. Accordingly, the steady state bias
factors of Johnson et al. (2002) and Vömel et al. (2020) should be decreased by subtracting 0.012. The resulting $S_S$ values
would then approach the $S_S$ values obtained here for SST0.1 and SST0.5, and better approximate the SST1.0 $S_S$ values.



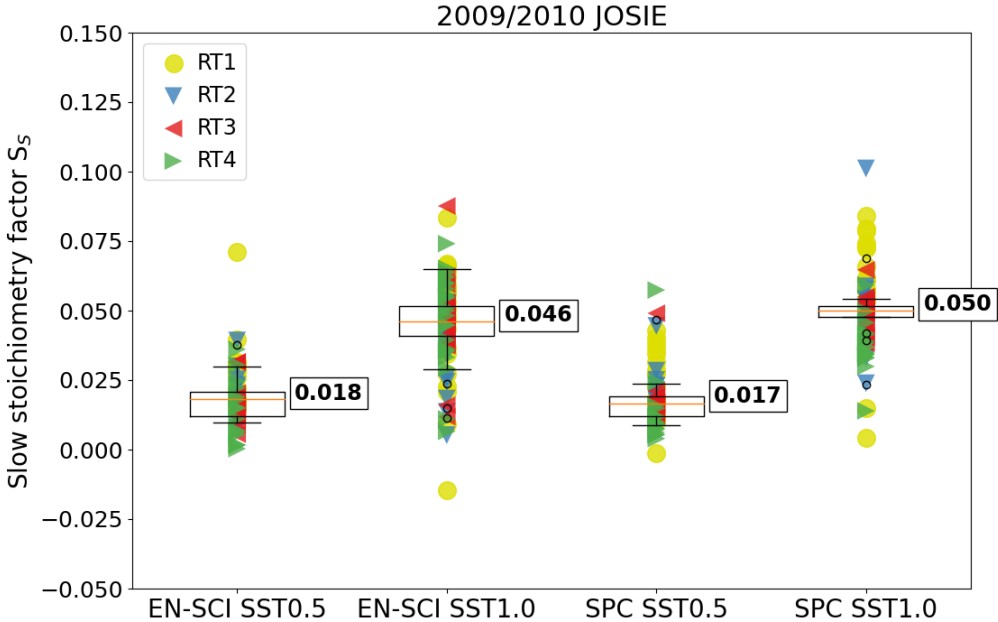


**Figure 3.** Whisker plots of the slow stoichiometry factor $S_S$ as the ratio of the measured $I_M$ minus $I_{B0}$ to the 25 min

convolved OPM current ($I_{OPM,C}$) obtained from JOSIE 2009 and 2010 for EN-SCI and SPC ozonesondes operated with the

SST0.5 and SST1.0. The yellow dots and triangle symbols (blue, red and green) represent the individual values obtained

from the four response tests RT1, RT2, RT3 and RT4, respectively. Thus, every ozonesonde profile is represented four times

in the graph. Whisker plots are represented by median plus the 25th and 75th percentiles (respectively, orange and black

horizontal lines for each instrument-SST combination).

Another difference between the new methodology and that of Vömel & Diaz (2010) is that we subtract $I_{B0}$ from the

ozonesonde signal prior to determining the stoichiometry. However, we also determined the $S_S$ values without correction of

$I_{B0}$; the results appear in Fig. S2 in Supplementary Material. It is noted that these $S_S$ values increase for all sensing solution

types by only 0.005-0.009. For SST0.5 and SST0.1, they approach the Vömel & Diaz (2010) values, but the substantially

lower $S_S$ values for SST1.0, as derived here (Table 2) cannot be explained exclusively by subtracting $I_{B0}$. Furthermore,

comparing Fig. 3 with Fig. S2, also demonstrates that the subtraction of the $I_{B0}$ value makes the determination of the $S_S$

values even more independent of the selected RT intervals, which is not the case without this prior subtraction (e.g. the RT1

values being significantly larger than the other RT values).

The factors reported by Johnson et al. (2002) and Vömel & Diaz (2010) are based on a limited sample of experiments (three

different sondes using three different solutions for a total of 22 runs in Vömel & Diaz, 2010) in contrast to the large

statistical sample in this study (Table 2). The difference between the two approaches – in terms of exposure to ozone or not -

may be then explained by assuming that when the overall excess stoichiometry originates from the secondary reaction

pathway, only half of it contributes to the slow cell current $I_S(t)$ and with the other half contributing to the fast cell current

$I_F(t)$. For SST05 and this SST1.0 this can be understood by the type of reaction mechanisms of the secondary reaction

pathway as proposed by Saltzman and Gilbert (1959): in this case, the extra stoichiometry caused by the buffer could be still





for 50% contributing to the relatively fast signal (R7) and 50% to the slow signal (R8) (see Appendix A). This would mean
that the stoichiometry of the secondary reaction pathway could be two times the stoichiometry factor $S_S$ of the slow ECC
current $I_S(t)$ determined here from the response tests RT1 to RT4 after $I_F(t) = 0$. However, for the $S_S$ values for the SST0.1,
even slightly larger than for SST0.5, explanations would be more speculative. More analysis and new JOSIE trials, for
example in the JOSIE simulation chamber, might be required to find the cause of varying factors among the different studies
and SSTs.

### 3.2.2 Initial Condition of Slow Current $I_S(t)$

With the derived $S_S$ values, the slow component of the sonde signal ($I_S$) is computed by convolution with the slow time
constant $\tau_s$= 25 min., as in Eq. (10) (brown line in Fig. 2). Note that, in practise, to determine $I_S(t)$, the measured current $I_M(t)$
minus $I_{B0}$ can be taken instead of the true generic ozone current $I_{OPM}(t)$, because their differences are rather small (less than
5-10%), at the same time the slow stoichiometry factors $S_S$ are also smaller than 0.1. From here on, we will use the measured
current $I_M(t)$ minus $I_{B0}$ to determine the slow current $I_S(t)$ along with the $S_S$ values listed in Table 2.

As Eq. (10) is a recursive expression, the initial conditions of $I_S$ reflect prior ozone exposure during pre-launch preparations,
although decaying exponentially in time. Exposure to ozone values during pre-launch will cause non-zero $I_S$ values at the
beginning of the simulation, impacting the boundary layer ozone profile (e.g., Fig. 10 in Vömel et al., 2020). Ideally, the
convolution of the slow component of the sonde signal is computed taking the pre-launch measurements into account. These
pre-launch measurements are available for JOSIE 2009/2010 (as in Fig. 4), but this is often not the case for operational
soundings. Using those JOSIE 2009/2010 pre-launch simulation data (with negative simulation times in Fig. 4), we found
that the best approximation of the true $I_S$ (red dashed line in Fig. 4, taking all the pre-launch measurements into account) is
obtained if $I_S(t_0)$ equals ($I_{B1}-I_{B0}$) multiplied with the exponential decay factor $X_S=Exp[-\Delta t/\tau_s]$, where $\Delta t$ is the time interval
between the measurement of $I_{B1}$ and the start of the launch (green dashed line in Fig. 4). It is important to mention here the
good agreement of the measured $I_{B1}$ value (yellow horizontal line in Fig. 4, subtracted by $I_{B0}$) with the convolved, pre-
launch, slow component $I_S$ (dashed red line) at $t = -2500$ seconds (time mark No.2 in Fig. 4). This reinforces the selection of
the $I_{B1} - I_{B0}$ measurement as a good pre-launch representation of the slow component of the ECC signal.

To apply this method in the ozonesonde network, it is essential to record the time difference between the $I_{B1}$ measurement
and the sonde launch. In GAW Report No. 268, the recording of the $I_{B1}$ timestamp is included in the SOP for ozonesonde
preparations. For the JOSIE 2009/2010 data, we will use this exponential decay method for the initial condition of the
convolved slow component at $t$=0. For the initial condition of the slow component $I_S(t_0)$ we investigated two other
alternatives:
•   $I_S(t_0) = I_{B1}-I_{B0}$, denoted by the horizontal yellow line in Fig. 4, which results in a slow component $I_S$ marked by the

purple solid line, which clearly overestimates the true $I_S$ in the beginning of the profile (up to about 3500 s).

•   $I_S(t_0) = 0$, for which the corresponding $I_S$, represented by the brown solid line in Fig. 4, underestimates the true $I_S$

up to about a simulation time of 2200s for the JOSIE 2009/2010 representative example here.

For stations with a time gap of several hours between the $I_{B1}$ measurement and the launch time, the current will have been
fallen back to the $I_{B0}$ (see the Uccle example in Fig. 1), resulting, after subtraction of $I_{B0}$, in this particular case $I_S(t_0) = 0$.

A better understanding of the ECC time response provided a justification for quality control indicators on the $I_{B0}$ ($< 0.03$ μA)
and $I_{B1}$ ($< 0.07$ μA) in GAW Report No. 268. In practice, often higher background currents $I_{B0}$ and $I_{B1}$ are recorded at the
sounding sites at the day of the launch. These high background currents are typically caused by the use of an inadequate gas



filter in the test unit, e.g. the filter provides ozone free air, but does not trap water vapour and contaminants in the laboratory
air that is filtered into the preparation equipment. A poor filter combined with a leaky photolysis cuvette producing ozone by
UV-photodissociation of oxygen with a Hg-discharge lamp can contaminate the air flow to produce in high background
current measurements. It appears that UV irradiation can produce substances that cause reactions similar to KI and $O_3$. There
are some indications (Newton et al., 2016) that high backgrounds may display be due to processes with 1/e-decay times ~ 25
min like the slow cell current $I_S$(t). Nevertheless, more research is necessary to investigate the cause and the time behaviour
of these high background currents in the course of the sounding in order to correct for this artifact properly. As stated by
ASOPOS 2.0 (WMO/GAW Report No. No. 268) the use of proper gas filters to provide ozone free, dry and purified air in
practice at the sounding site, is very essential in general, but also when applying the data processing proposed here.

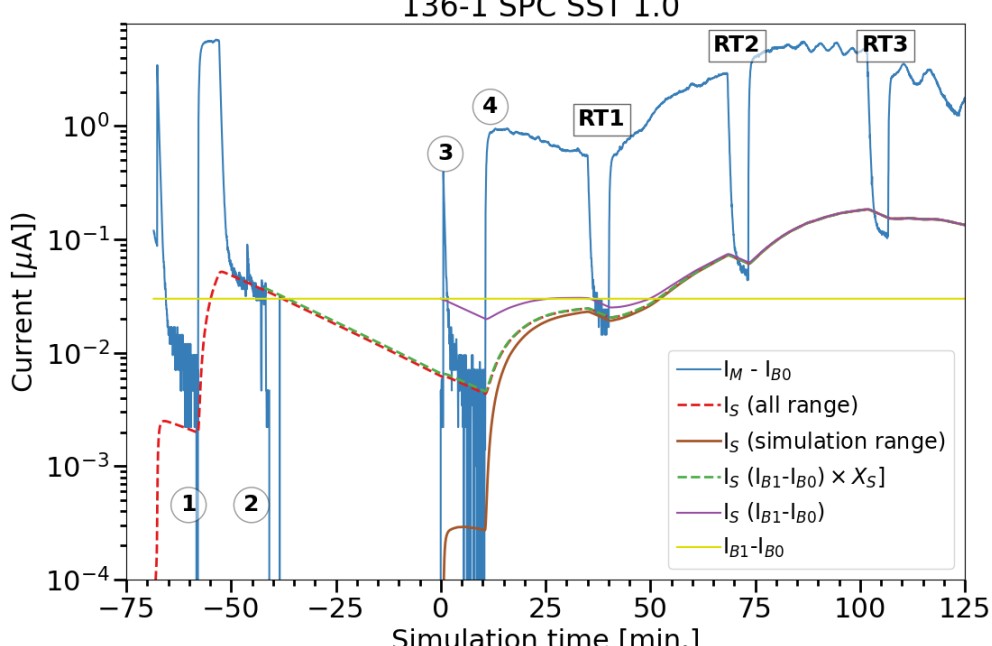


**Figure 4.** Convolved slow ECC current obtained from different initialization scenarios as function of the simulation time.


(details see text). The dashed red line is the convolved ECC current obtained from the measured $I_M$ minus $I_{B0}$, hereby
including all pre-launch measurements (with negative simulation times). Time stamps 1-4: 1= record $I_{B0}$; 2= record $I_{B1}$;
3=turn on pump motor (at simulation time t=0); 4= start ozone profile of simulation. RT1, RT2, RT3 are the first three in-
flight time response tests.

**3.3 Determination of the Fast ECC Ozone Sensor Current, $I_F$(t)**
After determining the slow component of the signal due to the secondary reaction pathway, we can subtract it from the
overall measured current $I_M$ - $I_{B0}$ to end up with the fast component $I_F$ (Eq. 7), as shown by the green line in Fig. 2. From the
fast component $I_F$(t), we can remove the time lag introduced by the 1/e time response of about 20-30 seconds through
deconvolution of $I_F$ (t) according to Eq. (12). In this paper, we use $\tau_F$ = 25 ± 4 seconds for EN-SCI, and $\tau_F$ = 21 ± 4 seconds
for SPC ozonesondes, which are the average fast time responses determined from all the simulation time response tests (RT1,



RT2, RT3, RT4) during JOSIE 2009/2010. The response times of the EN-SCI sondes are typically about 4 seconds larger
than the SPC-6A sondes due to the slightly lower pump flow rates and slightly larger volume of the cathode cell of the EN-
SCI sondes (Smit and Sträter, 2004a). In general, we found that the fast response times in upward as well in downward
direction agree within 1-2 seconds. Moreover, $\tau_F$ only varies marginally in flight with a slight decrease of less than 5-10 %
between the surface (RT1) and the upper part of the sounding (RT4). The in-flight $\tau_F$ values also agree very well with the $\tau_F$
values determined from the response tests made during the pre-flight preparation of the ECC sensor, which confirmed earlier
observations made during JOSIE (Smit and Sträter, 2004a). A close-up of the first time response interval RT1 is provided in
Fig. 5, in which also the deconvolved fast component is shown in yellow.

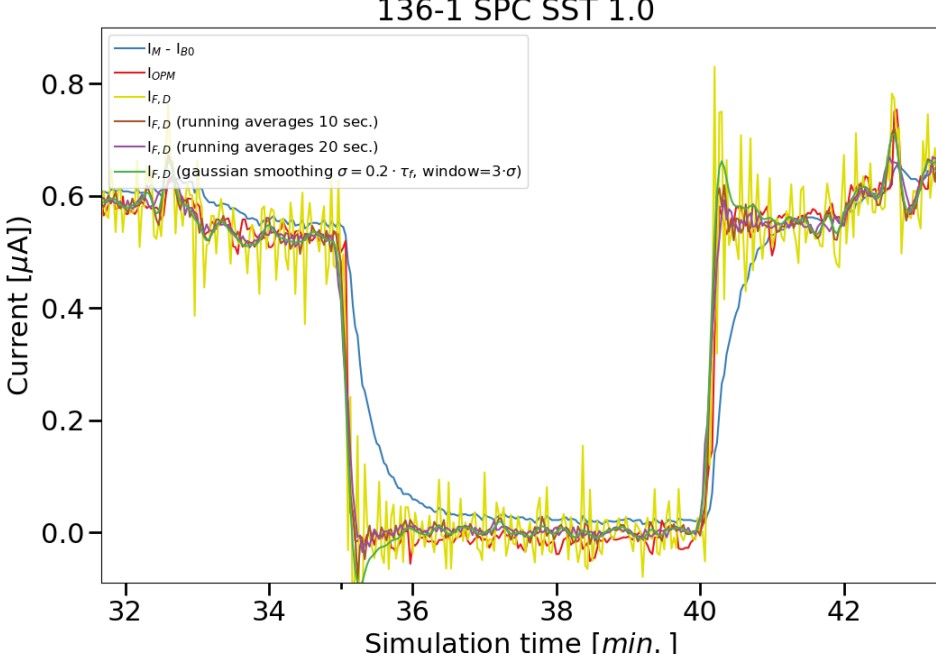


**Figure 5.** Example of a downward and upward response of a simulation run in the tropospheric part of the vertical profile to
show the impact of resolving the fast response effects on the measured cell current $I_M$ minus $I_{B0}$ ($I_M$-$I_{B0}$: blue solid line). The
fast, deconvolved current $I_{F,D}$, without smoothing, is shown in yellow, and with a moving average smoothing over a time
interval of 10 and 20s in brown and purple, respectively. The Gaussian smoothing applied on $I_{F,D}$ and used in this paper is
marked by the green line. For reference, the OPM current is shown in red.

Note that the deconvolution procedure introduces a substantial amount of noise in the data. To reduce this noise, the
deconvolved current signal should be smoothed. We therefore used a smoothing with a Gaussian filter with width equal to
20% of the time lag constant $\tau_F$ as in Vömel et al. (2020), their equations (10) and (11). Compared to other common
smoothing techniques, e.g. running averages with a time window of 10 seconds (see brown line in Fig. 5), this Gaussian
filter still has a slight phase shift with respect to the true signal ($I_{OPM}$, in red in Fig. 5), but outperforms other tested
smoothing algorithms in terms of reducing the noise level. The final smoothed deconvolved signal is shown in green in Fig.
5. It is obvious that, after correcting for the slow and the fast times responses in the signal, the resulting current better agrees
with the OPM current than the original measured current. It even exhibits small-scale features that are also present in the





fast(er) response OPM measurements. Small differences still remain that indicates the conversion efficiency, i.e.
stoichiometry of the fast reaction, slightly deviates from one.

**4. Comparison of Ozone Profiles Based on the Conventional Versus Updated Time Responses Correction Method**

To test the Time Responses Correction (abbreviated here as TRC) methodology as described in the previous chapter and a
first version in Vömel et al. (2020), we apply the methodology on individual ozonesonde profiles of the different JOSIE
simulations and compare those corrected profiles with the corresponding OPM measurements. This method involves the use
of the stoichiometry factors $S_S$ from Table 2 for the different ozonesonde-SST pairs and the application of the measured
pump efficiency factors of Nakano and Morofuji (2023) (Table 1). As opposed to this TRC method, ozone partial pressures
from profiles are determined according to the "conventional method", as recommended in ASOPOS (GAW Report No. 201;
GAW Report No. 268), e.g. using the constant background $I_{B1}$ correction with the Komhyr et al. (1986, 1995) efficiency
factors (Table 1). Both sets of processed profiles are compared to the OPM reference values which are enhanced by 1.23%
compared to earlier JOSIE publications due to the newly revised ozone absorption cross-section at 254 nm wavelength
(Hodges et al., 2019). In 2024-2025 the new cross-section will be introduced into the global ozone observation networks
using UV-photometry (BIPM, 2022). The comparisons are made for two different JOSIE campaigns: (i) JOSIE 2009/2010
with mid-latitude profiles and well-established ozonesonde preparation procedures, and (ii) the JOSIE 2017 campaign with
mostly tropical profiles and good ozonesonde preparation procedures.
All here presented comparisons of the TRC with the conventional method are processed as function of flight time. However,
to present the results as vertical profiles, they are mapped on a pressure grid with successive pressure levels of $P_i$=0.98 x $P_{i-1}$
between 1000 and 5-6 hPa. Hereby, all presented JOSIE experiments are based on a pressure, temperature and ozone profile
simulating a balloon ascent velocity of about 5 m/s, such that a quasi-realistic linking between the simulated flight time and
pressure scale is obtained.

**4.1 Ozone Profiles from JOSIE 2009-2010 for SST1.0 and SST0.5**

In Figure 6, the relative differences with the OPM for the conventionally (left diagrams) and TRC (right diagrams) processed
ozonesonde profiles of JOSIE 2009/2010, respectively, are shown for each pair of sonde (SPC6A or EN-SCI) and solution
type (SST0.5 or SST1.0), respectively, including the mean (black solid lines) and its 1σ-standard deviation. The absolute
ozone partial pressure differences are presented in the supplementary material (Fig. S3).



**Figure 6** JOSIE 2009/2010: Relative differences with the OPM for the conventional (left diagrams) and TRC (right diagrams) processed ozonesonde profiles for four pairs of sonde type and SST shown as scatter plots in four different colors in the panels a-d: SPC6A/SST1.0 (a: blue dots), EN-SCI/SST0.5 (b: red dots), SPC6A/SST0.5 (c: green dots), and EN-SCI/SST1.0 (d: brown dots), respectively. In each diagram for both methods the mean and 1σ-standard deviation of the relative differences are included (solid black line). The black dashed lines in the TRC-diagrams are the linear regressions of the difference of the ozonesonde to the OPM as function of the pressure (on a logarithmic scale). A summary plot is provided in Fig. S4, and absolute differences are available in Fig. S3 of the Supplementary material.

For the conventional method, large relative deviations from the OPM exist in the pressure intervals response-time tests (in particular RT1, RT2, RT3) take place during a simulation. This can be explained by the difference in response time between the OPM and the ozonesondes and the fact that when ozone concentrations are close to zero, the relative differences will be



magnified. The TRC method is able to correct well for the time response differences, as illustrated by the small relative
differences, although with higher uncertainty (1σ-standard deviation) compared to adjacent pressure levels.  A major
improvement of the TRC methodology compared to the conventional corrections is the fact that the relative differences with
respect to the OPM are almost pressure-independent, hence past ozone exposures. Up to about 13 hPa (Z≈30 km), only a
slightly increasing bias with decreasing pressure exists between the overall mean of the TRC-corrected ozonesondes and
OPM for the JOSIE 2009/2010 sample (black dashed linear regression lines in Fig. 6).

At pressures lower than 13 hPa the SPC sondes exhibit a declining behaviour, which is discussed in the next section. Overall,
both EN-SCI SST0.5 and SPC SST1.0 agree very well within a few percent, with the TRC methodology using the correct
pump efficiencies (see also Fig. S4). Consistent with earlier JOSIE and BESOS campaigns (Smit et al., 2007; Deshler et al.,
2008), for both sonde types, SST0.5 gives around 3-5% lower ozonesonde readings than SST1.0, whereas, for both SSTs,
SPC ozonesondes read ~ 3-5% lower than EN-SCI.

**4.2 Ozone Profiles from JOSIE 2017 for SST1.0, SST0.5, and SST0.1**
During the JOSIE 2017 campaign, tropical ozone profiles were simulated for three different SSTs: SST1.0, SST0.5 and
SST0.1 (Thompson et al., 2019). No time-response tests were performed during these simulations. Therefore, for SST1.0 and
SST0.5, the stoichiometry factors, $S_S$, derived from the JOSIE2009/2010 data have been applied. However, the SST0.1
solution was not tested during the JOSIE 2009/2010 campaign. Therefore, for this SST, we determined the stoichiometry
factors $S_S$ with the same method as described in Sect. 3.2.1, but with time-response tests during ozonesonde laboratory
measurements with a calibrated ozone analyser (details in Appendix B). The derived $S_S$ factor is 0.023 ± 0.005. For the
JOSIE 2017 campaign data, the initial value of the slow current component $I_S$ at the start of the simulation at $t$=0 (Sect.
3.2.2) has been chosen to equal 0 (i.e. equal to $I_{B0}$ before subtracting $I_{B0}$), as there were usually a few hours between the end
of the day of launch preparations and the start of the simulation, such that $I_{B1}$ has decayed to $I_{B0}$. The Uccle experiments
(Fig. 1) illustrated that the measured current with a zero-air source falls after two hours back to the $I_{B0}$ value measured at the
beginning of the preparation.

The differences of the JOSIE 2017 ozonesonde profiles with the corresponding OPM profile using the conventional and TRC
data processing methodologies are shown in Figure 7; the absolute differences appear in Fig. S5. The most prominent feature
for the conventional corrections, sonde type-SST combinations, is the dependence of the OPM differences on pressure or
measured ozone amounts: the mean relative differences are largest (as well as the corresponding standard deviations) just
below the tropopause at ~200 hPa, where the ozone partial pressures are minimal; the mean relative differences increase with
decreasing pressure in both troposphere and stratosphere (also obvious in Fig. S6). The conventional method deviates strong
from the OPM in the upper troposphere at about 100 hPa for the tropical ECC ozone profiles. In contrast, when the TRC-
method is applied to the data, the pressure/ozone amount dependence of the relative difference almost completely disappears.
For the standard EN-SCI/SST0.5 and SPC/SST1.0, there remains a slightly increasing bias with decreasing pressure (black
dashed lines), while for the SST0.1 ozonesonde simulations, there is a tendency for decreasing (negative) relative differences
with decreasing pressure. For both SPC and EN-SCI, SST0.1 ozone measures about 10% lower than OPM in the
stratosphere, compared to the recommended SOP (SPC/SST1.0 and EN-SCI/SST0.5).



JOSIE 2017 SPC/SST1.0     JOSIE 2017 EN-SCI/SST0.5

(a)                   (b)

JOSIE 2017 SPC/SST0.1     JOSIE 2017 EN-SCI/SST0.1

(c)                   (d)

**Figure 7.** JOSIE 2017: Differences with the OPM for the conventionally (left) and TRC (right) processed ozonesonde
profiles for the four sonde-SST pairs as scatter plots: SPC6A/SST1.0 (a: blue dots), EN-SCI/SST0.5 (b: red dots),
SPC6A/SST0.1 (c: purple dots), EN-SCI/SST0.1 (d: yellow dots). In each diagram for both methods, mean and 1$\sigma$-standard
deviations are solid black lines. The black dashed lines in the TRC-diagrams are the linear regressions of the sonde-OPM
differences as a function of the pressure on a logarithmic scale. A summary plot appears in Fig. S6 and absolute differences
are in Fig. S5 of Supplementary material.

When comparing the mean relative OPM offsets after processing the ozonesonde measurements with TRC methodology for
the two JOSIE campaigns, i.e. Figs. 6 and 7 (also in Figs. S4 and S6), we note that the network standards SPC/SST1.0 and
EN-SCI/SST05 are a few percent larger in the stratosphere for the "tropical" JOSIE 2017 campaign. That is, those mean
relative differences are manifest in both cases as a slightly decreasing relative bias with increasing pressure during both



campaigns. These differences are independent of post-ozone exposure and profile type (mid-latitude or tropical), in contrast
to the conventional methodology which exhibits this past ozone memory effect. A striking disagreement between the profile-
OPM offsets between JOSIE 2009/2010 and 2017 occurs at the lowest pressure range, lower than ~13 hPa. For the JOSIE
2009/2010 data, the mean relative differences with the OPM display a stronger pressure dependence in this lowest pressure
range, distinctly different for both sonde types, in contrast to the JOSIE 2017 mean relative OPM differences. The origin of
this different behaviour above 13 hPa lies most likely in pump temperature differences between the simulated profiles.
Whereas the mean pump temperature is close to 21°C in this pressure range in JOSIE 2009/2010, it is near 15°C for the
tropical profiles in JOSIE 2017. Simultaneous temperature measurements during JOSIE 2017 revealed that the cell
temperatures are about 5 to 10°C lower than the corresponding pump temperatures, depending on the sonde type.
Specifically the differences between pump and cell temperature are more at the high end range of this temperature interval
for EN-SCI sondes, and at the low end range for the SPC due to differences in thermal contact between cells and pump. With
these cell temperatures and taking the boiling temperatures at those low pressures into account, it turns out that the solutions
in the SPC sondes tested in JOSIE 2009/2010 may already start boiling at higher ambient air pressures than during JOSIE
2017. Cell weights were measured before and after all simulations for both campaigns. The weight loss due to
evaporation/boiling of the sensing solution was considerably higher during JOSIE 2009/2010 than in JOSIE 2017: about a
factor 2 for EN-SCI/SST0.5 and even a factor 3 for SPC/SST1.0. Although at these reduced ambient air pressures the
absorption efficiency is not critical (Tarasick et al., 2021), the sensing solutions losses of the sondes may have become so
large during JOSIE 2009/2010 that the absorption efficiency has been declined. This may explain the underestimation of the
ozone concentrations at low pressures for the JOSIE 2009/2010 profile simulations, in particular for SPC ozonesondes.

### 4.3 Differences Between Different Pairs of Sonde Type and SST

For each pair of ozone sonde type and SST for JOSIE 2009/2010, JOSIE 2017 and combined JOSIE 2009/2010 and 2017
(for SPC/SST1.0 and EN-SCI/SST0.5) a linear regression has been calculated as a function of pressure on a logarithmic
scale for the TRC sonde-OPM relative differences within ±30% for pressures up to 13 hPa. These linear regression lines are
shown in Figs. 6 and 7 as black dashed curves in the TRC diagrams for the different sonde-SST pairs; they agree well with
the corresponding averages (black solid lines in TRC diagrams). All TRC-sonde/SST pair relative difference scatterplots
display variations within 3-7% with altitude between the surface at P=1000 hPa and the upper end of the profile at P=10 hPa,
as can be seen in Table 3 that displays the relative sonde-OPM differences at the intercepts P=1000 hPa and P=10 hPa of the
linear regression. Table 3 illustrates the same typical differences of 3-5% for the same sonde type but different SST1.0 or
SST0.5, as first observed in JOSIE 2000 (Smit et al., 2007). Figures S4 (a & b) and S6 (a and b) show the persistence of
these systematic differences in detail for the conventional and TRC method as function of pressure (i.e. altitude). The low
buffered (SST0.1) EN-SCI or SPC-6A sondes slightly underestimate ozone by a few percent compared to the OPM. It is
noteworthy that the EN-SCI/SST0.1 OPM offsets decrease over the course of the sounding, in contrast to all other sonde-
SST pairs for which the relative differences increase (Table 3: last column).





**Table 3.** Relative differences of the sonde to the OPM at the P= 1000 hPa and P=10 hPa intercepts of the linear regression as a function of Log$_{10}$(P) obtained from the different JOSIE data sets (Figs. 6-7) and for the sonde pairs SPC-6A and EN-SCI with different sensing solutions SST1.0, STT0.5 and SST0.1. Included are also the relative differences between EN-SCI and SPC6A sondes when operated at the same SST (last three rows).

| Data set | Number of Samples | Rel. Differences in % Sonde to OPM at intercept P=1000 hPa | Rel. Differences in % Sonde to OPM at intercept P=10 hPa | Rel. Differences in % Sonde to OPM between P is 1000 and 10 hPa |
|---|---|---|---|---|
| | SPC-6A /SST1.0 | | | |
| JOSIE 2009/2010 | 23 | 1.69 | 5.47 | 3.8 |
| JOSIE 2017 | 11 | 3.12 | 7.68 | 4.6 |
| JOSIE 2009/2010 + 2017 | 34 | 2.26 | 6.44 | 4.2 |
| | SPC-6A /SST0.5 | | | |
| JOSIE 2009/2010 | 20 | -2.0 | 3.62 | 5.6 |
| | SPC-6A /SST0.1 | | | |
| JOSIE 2017 | 6 | -3.52 | -2.24 | 1.8 |
| | EN-SCI /SST1.0 | | | |
| JOSIE 2009/2010 | 25 | 3.89 | 11.26 | 7.4 |
| | EN-SCI /SST0.5 | | | |
| JOSIE 2009/2010 | 15 | 1.35 | 8.30 | 7.0 |
| JOSIE 2017 | 20 | 1.93 | 6.21 | 4.3 |
| JOSIE 2009/2010 + 2017 | 35 | 1.72 | 7.02 | 5.3 |
| | ENSCI /SST0.1 | | | |
| JOSIE 2017 | 20 | 0.35 | -2.27 | -2.6 |
| | | | | |
| SST | EN-SCI – SPC6A | | | |
| SST1.0 | | 1.63 | 4.82 | 3.2 |
| SST0.5 | | 3.92 | 3.40 | -0.5 |
| SST0.1 | | 3.87 | 0.03 | -3.4 |





**5  Conversion Efficiency of TRC Method Calibrated to OPM**
In the previous section it was shown that the TRC-method resolves the dependence of the measured ozonesonde profile from
the past exposure of ozone, whereas the deconvolution of the remaining fast ozone sensor current resolves effectively the
impact of gradients in the profile caused by the 20-30 sec time response of the ECC-sensor. The sonde to OPM comparisons
presented in section 4 for the mid-latitude profiles of JOSIE 2009/2010 (Fig. 6) and tropical profiles of JOSIE 2017 (Fig. 7)
demonstrate that the TRC results are independent of the shape of the simulated ozone profiles, in contrast to the results
obtained by the conventional method (e.g. Smit et al., 2007; Deshler et al., 2008, 2017; Thompson et al., 2019). Further, the
TRC results show a strong consistency of the mean relative differences with the OPM for the different sonde types-SST
combinations across the different (grouped) JOSIE campaigns (see also Figs. S4 and S6). Therefore, those relative mean
differences can be characterized by the linear regression curves as a function of $\mathrm{Log}^{10}(P)$ in Figs 6-7 and directly linked to
the OPM. As such, these linear regression lines (hereafter referred to as "calibration curves") could be applied as the final
correction step of the TRC methodology, tracing the ozonesonde measurements back to the OPM as the reference
instrument.

**5.1 Parameterisation of the Overall Conversion Efficiency $\eta_C$**
The linear regressions of the relative differences of the sonde to the OPM (Figs. 6-7) of the TRC method can be interpreted
as the correction term of the overall conversion efficiency $\eta_C$ when deviating from one for each of the different pairs of
sonde type and SST. The overall conversion efficiency $\eta_C$ in Eq. (6) can be expressed as a function of the ambient air
pressure of the vertical sounding:
$\eta_C(P) = 1 + F_C(P)$ (17)
where $F_C(P)$ is the so-called correctional term of $\eta_C$ as a function of the ambient air pressure P, which is parameterised by the
linear regression fit of the relative sonde-OPM deviations as a function of $\mathrm{Log}_{10}(P)$ and substituted in Eq. (17). This means
that the overall conversion efficiency $\eta_C(P)$, calibrated to the OPM, has the following parameterisation
$\eta_C(P) = 1 + a + b \cdot \mathrm{Log}_{10}(P)$ (18)
The linear regression curves derived for the different pairs of SPC-6A, EN-SCI with SST1.0, SST0.5, or SST0.1 obtained
for the different JOSIE campaigns are shown in the TRC diagrams of Figs. 6-7 by the dashed black line. From Fig. 6-7 and
Table 3, it is obvious that the relative OPM offsets (and the resulting linear regressions) for the same pairs of sonde and
SST05 or SST1.0 are very similar in JOSIE 2009/2010 and JOSIE 2017. Thus, to achieve the best statistics, the results for
those campaigns are lumped together in Fig. 8.

The results of the parameterisation of $\eta_C(P)$, i.e. the offset a and the slope b (Eq.18), including their uncertainties $\Delta a$ and the
slope $\Delta b$, respectively, are listed for the different pairs of sonde type and SSTs as JOSIE (2009/2010 + 2017) in Table 4. The
sonde/SST pairs operated with SST0.5 and SST1.0 cover mid-latitude as well as tropical ozone profile conditions, i.e. the
resulting $\eta_C(P)$ functions are independent of the ozone profile. Based on this, we expect that the $\eta_C(P)$ for the SST0.1, which
could only be derived in this study for the tropical JOSIE-2017 conditions, can also be applied to non-tropical ozone profiles.
Likewise, we expect that the $\eta_C(P)$ determined from JOSIE 2009 only for the SPC/SST0.5 and EN-SCI/SST1.0 pairs are
valid for tropical ozone profiles.

The calibration functions are presented here (Table 4) as a function of pressure, but this does not mean that they are really
pressure-dependent. However, the goal is to provide a practical empirical representation of the overall performance of the
ozonesonde, ascending with a balloon at ~5m/s. The calibration functions can thus be interpreted as the correction term of



the overall conversion efficiency of the ECC sonde when deviating from one, but the origins are still uncertain. Most likely
the term relates to the unknown stoichiometry of the fast chemical reactions converting ozone into free iodine, in other
words, the fast ECC current $I_F$. This is supported by the shape of the vertical profiles of the absolute $P_{O_3}$-differences of the
ECC sonde compared to the OPM for the TRC, shown for the JOSIE 2009/2010, JOSIE 2017 and for the JOSIE 1996-2002
data (described in section 5.2), in the middle diagrams of Figures S3, S5 and S7, respectively. Indeed, in the middle
stratosphere, the shapes of the residual currents compared to the OPM are more or less in phase with the simulated ozone
profiles. This is most pronounced for the JOSIE-2017 tropical profiles (Fig. S5) and might indicate that these residual
currents result from the fast chemical conversion and not from the 25-min delayed slow reaction. In the latter case, a phase
shift between the residual currents and the ozone profile would be expected. The observed increase with altitude of typical 3-
7% in the calibration functions (Tables 3 & 4) might derive from a small slightly increasing change stoichiometry of the fast
$O_3$ conversion due to an increase of KI concentration and buffer strength caused by evaporation during the sounding.

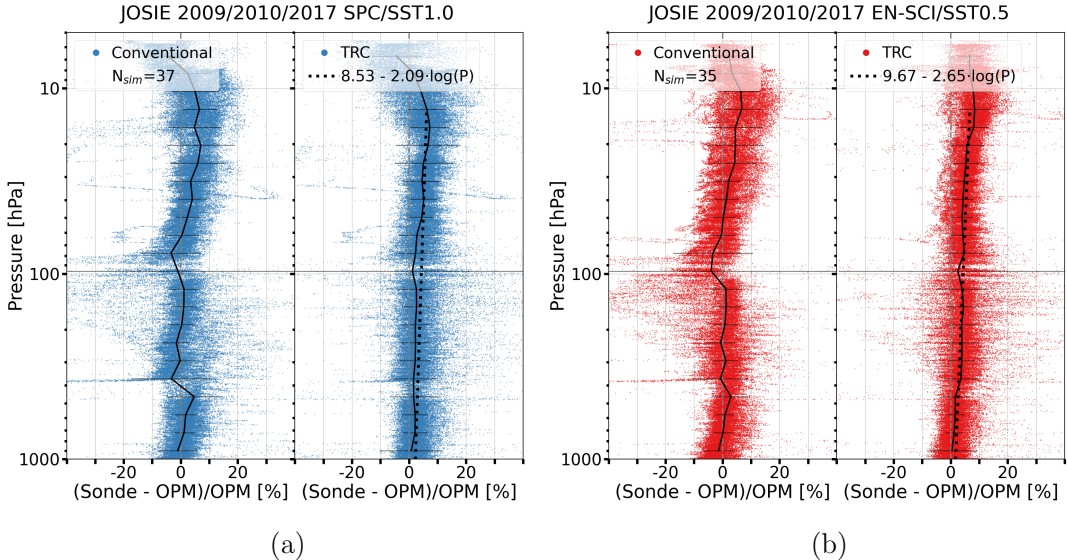

(a)                                                    (b)


**Figure 8.** JOSIE 2009/2010 and 2017: Relative differences with the OPM for the conventional (left diagrams) and TRC
(right diagrams) processed ozonesonde profiles for two pairs of sonde type and SST shown as scatterplots for
SPC6A/SST1.0 (a: blue dots) and EN-SCI/SST0.5 (b: red dots), respectively. In each diagram for both methods the mean
and 1σ-standard deviation are included (solid black line). The black dashed lines in the TRC-diagrams are the linear
regressions of the differences of the ozonesonde to the OPM as function of the pressure (on a logarithmic scale).

Although the cell temperatures of the ozonesondes (both SPC6A/SST1.0 and EN-SCI/SST0.5) in JOSIE. 2009/2010 were
about 10 ºC higher than in JOSIE 2017 there are no direct indications that there is any cell temperature dependence of the
calibration functions. This is demonstrated by the fact that SPC6A/SST1.0 and EN-SCI/SST0.5 for both campaigns show
very similar OPM deviations over the course of the sounding when compared at the intercept points at P=1000 and 10 hPa
(Table 3). However, temperature dependence cannot be completely excluded, in as much as the chemical reactions involved
in the KI+$O_3$ chemistry may have significant temperature dependencies. Again, further in-depth investigations are needed.




**Table 4.** Parameterisation (offset a and slope b) of the calibrated conversion efficiency $\eta_C$ (P) (Eq. 18) for the different pairs
of SPC-6A or ENSCI with SST1.0, SST0.5, or SST01 derived from the results of JOSIE 2009/2010 and JOSIE 2017.
Included are the uncertainties $\Delta a$ and $\Delta b$ of the offset a and slope b in Eq. 15, respectively. The parameterisation of $\eta_C$ (P)
is valid from P=1000 hPa until P=13 hPa (Z≈30 km) for SPC, and for EN-SCI to 10 hPa (Z≈32-33km).

| Sonde Type / SST | Number of Samples | TRC-Conversion Efficiency $\eta_C(P) = 1 + a + b \cdot Log_{10}(P)$ Eq. (18) | | JOSIE Data Set |
|---|---|---|---|---|
| | | Offset $a \pm \Delta a$ | Slope $b \pm \Delta b$ | |
| SPC-6A /SST1.0 | 34 | $(8.53 \pm 0.07) \times 10^{-2}$ | $(-2.09 \pm 0.03) \times 10^{-2}$ | JOSIE (2009/2010 + 2017) |
| SPC-6A /SST0.5 | 20 | $(6.43 \pm 0.08) \times 10^{-2}$ | $(-2.81 \pm 0.04) \times 10^{-2}$ | JOSIE 2009 |
| SPC-6A /SST0.1 | 6 | $(-1.60 \pm 0.12) \times 10^{-2}$ | $(-0.64 \pm 0.05) \times 10^{-2}$ | JOSIE 2017 |
| EN-SCI /SST1.0 | 25 | $(14.94 \pm 0.07) \times 10^{-2}$ | $(-3.68 \pm 0.03) \times 10^{-2}$ | JOSIE 2009 |
| EN-SCI /SST0.5 | 35 | $(9.67 \pm 0.06) \times 10^{-2}$ | $(-2.65 \pm 0.03) \times 10^{-2}$ | JOSIE (2009/2010 + 2017) |
| EN-SCI /SST0.1 | 20 | $(-3.58 \pm 0.09) \times 10^{-2}$ | $(1.31 \pm 0.04) \times 10^{-2}$ | JOSIE 2017 |



**5.2 Application to JOSIE 1996 + 1998 + 2000 + 2002 data**
The calibrated $\eta_C$(P) functions derived from JOSIE 2009/2010 and JOSIE 2017 (Table 4) for the different sonde/SST pairs
are applied to TRC processed ozonesonde data of JOSIE 1996 + 1998 + 2000 + 2002, in Figure 9, again as relative
differences to the OPM. The JOSIE 1996 + 1998 + 2000 datasets and results were described in detail by Smit and Kley
(1998) and Smit and Sträter (2004a, 2004b) and analysed by Smit et al. (2007). For JOSIE 1996, we excluded data from
NOAA and CNRS because their operating procedures deviated too greatly from the Komhyr (1986) procedures; JOSIE 2002
was a small campaign in which only 3 simulation runs were made with 10 SPC/SST1.0 sondes. The setup of the earlier
campaigns was similar to the JOSIE 2009/2010 or JOSIE 2017 experiments. In the earlier campaigns mostly mid-latitude
ozone profiles were simulated with the same four combinations of EN-SCI or SPC with either SST0.5 or SST1.0 (although
the sample sizes with SST0.5 were rather small). The largest difference between JOSIE 2009/2010 and the early JOSIE
campaigns lies in the preparation of the ozonesondes: in JOSIE 2009/2010, the same SOPs were followed by the three
operators; ozonesondes "flown" in the earlier JOSIE-campaigns being prepared by different teams of people with a variety
of SOPs.

The comparisons with the OPM in Fig.9 are displayed for the TRC results, once not calibrated ($\eta_C$(P) = 1.00, middle
diagrams) and once calibrated ($\eta_C$(P) from Table 4, right diagrams), while the results for the conventional method (left
diagrams) are included. From the figure it is obvious that independent of the sonde type (SPC-6A or EN-SCI) or sensing
solution type (SST1.0, SST0.5), after applying $\eta_C$(P) the residual average curves (black solid lines) are within less than ±1%
deviation from the "zero" over the entire vertical profile until 7-10 hPa. This means that with the TRC, combined with the
use of the specific $\eta_C$(P) for the various sonde-SST pairs, there are no longer systematic bias effects in the measured vertical
ozonesonde profiles with respect to the OPM as a function of pressure (i.e. altitude). The use of the calibrated TRC can be a
powerful tool to homogenize long term ozone records in the global ozonesonde network, so that these are now traceable to
one reference standard, i.e. the OPM at the WCCOS. The application of the TRC with the use of the calibration functions on
the JOSIE 2009/2010 and JOSIE 2017 datasets is also illustrated in the figures S3, and S5 in the Supplementary Material,




TRC + Calibration. This information is also provided for the absolute differences for the early JOSIE campaigns in Fig. S7.

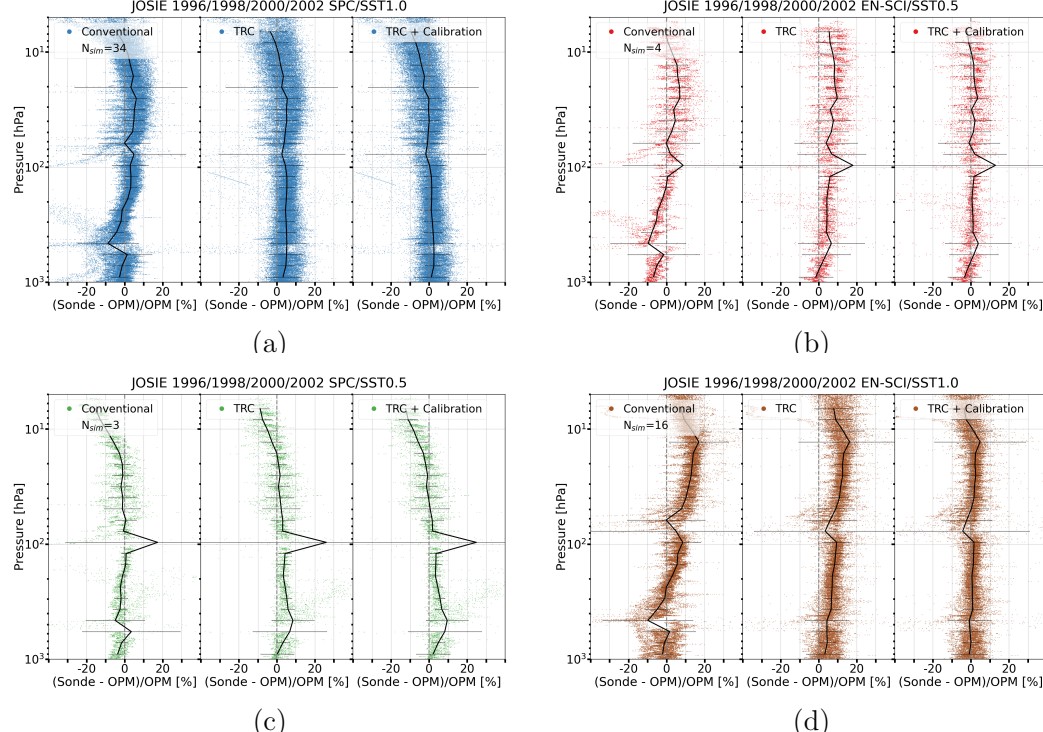



**Figure 9.** JOSIE 1996 +1998 + 2000 + 2002: Relative differences [%] with the OPM for the "conventional" (left diagrams
of panels a-d), "TRC" (middle diagrams of panels a-d), and "TRC + application of calibration functions" (right diagrams of
panels a-d) processed ozonesonde profiles for four pairs of sonde type and SST, shown as scatter plots in four different
colors in the panels a-d: SPC6A/SST1.0 (a: blue dots), EN-SCI/SST0.5 (b: red dots), SPC6A/SST0.5 (c: green dots), and
EN-SCI/SST1.0 (d: brown dots), respectively. In each diagram for both methods the mean and 1σ-standard deviation of the
relative differences are included (solid black line). The absolute difference plots are available in the Supplementary Material
(Fig. S7), and a summary plot of the relative differences in Fig. S8.




**6. Contribution Individual Correction Steps and Uncertainty Budget of the TRC Method**
In this section we quantify the impact of the individual corrections made in the TRC method and estimate their uncertainty
contributions to the overall uncertainty of the ozone partial pressure derived from the measured ECC-ozone sensor current.

**6.1 Contribution of Correction Steps of TRC-Method for Mid-Latitude and Tropical Conditions**
To derive from the measured cell current $I_M$ the partial ozone pressure in the ambient air the TRC method includes four
different corrections: (i) constant background current $I_{B0}$; (ii) slow cell current $I_S$; (iii) time lag of fast current $I_F$: deconvolved
fast cell current (incl. smoothing); (iv) new pump efficiency (Nakano et al., 2023). The impact of the different corrections on
the measured cell current as a function of pressure (i.e. $Log^{10}$ (P)) are shown in Figure 10 for mid-latitude (JOSIE
2009/2010) and tropical (JOSIE 2017) vertical profile conditions for the standard sonde type –SST pairs, SPC6A/SST1.0 and
EN-SCI/SST0.5, respectively.

A first, obvious, observation to make is that the corrections for a decreasing pump efficiency are for all sonde type- SST
pairs identical and at pressures smaller than 100 hPa increase slowly but significantly from 1 % at P=100 hPa to 12% at P =
10 hPa and to almost 20 % at P = 5 hPa. In the upper part of the profile (above 25 hPa) it is the dominating correction. In the
lower part, below 100 hPa , the constant background $I_{B0}$ (brown line) and the past ozone dependent slow cell current $I_s$
(yellow line) are the major corrections, particularly in the upper tropical troposphere, with its very low ozone concentrations
(panels E and F). Here, those corrections can amount up to about 10-15%, depending on e.g. the amplitude of the measured
$I_{B0}$ values. In this context, we also note that, because of the larger $S_S$ values for SPC6A/SST1.0, the past ozone dependent
slow current ($I_S$) correction will be about a factor 2 larger than the $I_S$ correction for the ENSCI/SST0.5, in all diagrams of
Fig. 10. On top of this effect, for SPC6A/SST1.0 JOSIE 2009/2010 (diagram b in Fig. 10), above 10 hPa, the relative $I_S$
correction is even rapidly increasing in absolute value due to the limited performance of the SPC6A sonde due to substantial
losses of the sensing solution caused by boiling effects, as explained before in section 4.2. The impact of the time lag
correction of the fast current is of the order of $\pm$ 5 % , and of course strongly dependent on the local vertical ozone gradient.
Therefore, it can even become the dominant correction in the tropical UTLS region (between 5-10%), with its strong vertical
ozone gradient (diagrams e-f). Finally, we mention that very similar results are obtained for the ozonesonde types combined
with SST0.1, which are shown in the supplementary material (Fig. S9).



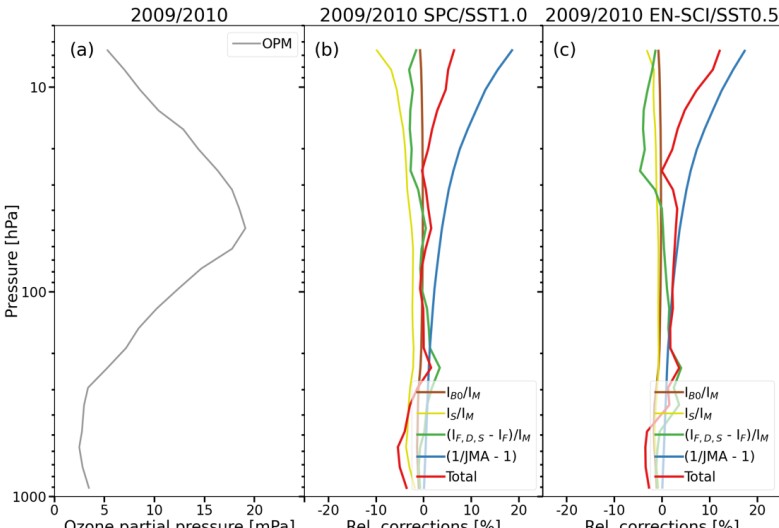


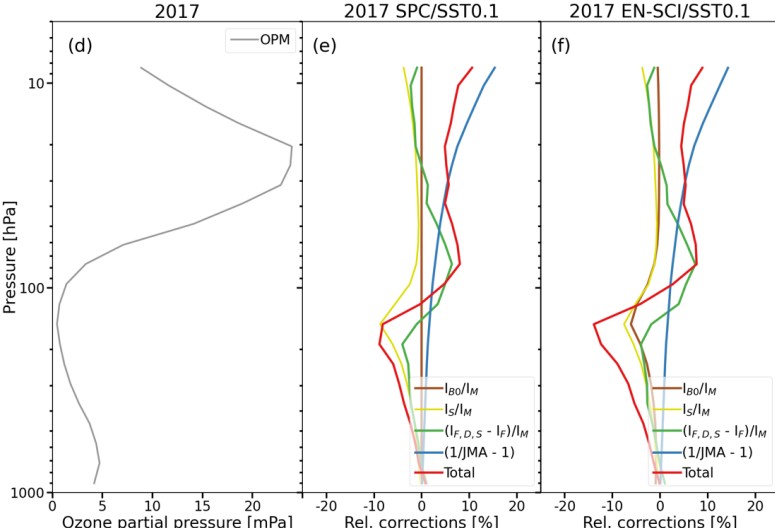


**Figure 10.** Relative corrections of TRC method for typical mid-latitude (upper diagrams A, B, C: JOSIE 2009/2010) and
tropical (diagrams d, e, f: JOSIE 2017) ozonesonde profiles, respectively, showing the influence of the different correction
steps for the new TRC method for SPC SST1.0 (diagrams b and e) and EN-SCI SST0.5 (diagrams c and f). The total
correction (red line) consists of: (i) $I_{B0}$ (brown line); (ii) $I_S$ (yellow/green line); (iii) De-convolution $I_F$ (green line); (iv) Pump
efficiency (blue line: Nakano et al., 2023).





### 6.2 Uncertainty Budget of the TRC Method


For the conventional method a detailed uncertainty budget has been studied by Tarasick et al. (2021) and described in detail
in the GAW 268 Report (Eq. E-3-1), together with practical guidelines to determine the overall uncertainty from the
individual instrumental and procedural contributions. It is assumed that the uncertainties are random, uncorrelated, and
normally distributed and following Gaussian statistics. In case of the TRC, the overall relative uncertainty of $P_{O3}$ is derived
from Eq. (7), which has slightly changed compared to formula E-3-1 in GAW#268 (2021) as follows:
$$\frac{\Delta P_{O3}}{P_{O3}} = \sqrt{\left(\frac{\Delta \eta_P}{\eta_P}\right)^2 + \left(\frac{\Delta \eta_A}{\eta_A}\right)^2 + \left(\frac{\Delta \eta_C}{\eta_C}\right)^2 + \frac{(\Delta I_F)^2}{(I_F)^2} + \left(\frac{\Delta T_P}{T_P}\right)^2 + \left(\frac{\Delta \Phi_{P0}}{\Phi_{P0}}\right)^2 + \sum \varepsilon_i^2} \qquad (19)$$

The additional term $\varepsilon_i$ represents additional random uncertainties (Tarasick et al., 2021); in case of the TRC these can be e.g.
the relative uncertainty contributions by the used numerical schemes of either the convolution to obtain $I_S(t)$ or the
deconvolution of $I_F(t)$ and its additional smoothing.
To determine the uncertainty budget for TRC in Eq. (19) the uncertainty contributions $\Delta \eta_P$, $\Delta \eta_A$, $\Delta I_M$, $\Delta I_{B0}$, $\Delta T_P$, and $\Delta \Phi_{P0}$
are exactly the same as in GAW Report No. 268 (2021) following the guidelines in its Annex-C. However, the recipes to
determine the uncertainty contributions of the time varying $I_F(t)$, and the pressure dependent $\eta_C(P)$ (See Table 4) differ from
GAW#268:

*Uncertainty contribution $\Delta I_F$:*
From Eq. (7) the relative uncertainty of the fast sensor current $I_F(t)$ can be derived:
$$\frac{\Delta I_F}{I_F} = \sqrt{\frac{(\Delta I_M)^2 + (\Delta I_{B0})^2 + (\Delta I_S)^2}{(I_M - I_{B0} - I_S)^2}} \qquad (20)$$

Here $\Delta I_{B0} \approx 0.01$ μA, obtained from the $I_{B0}$ time series from Uccle. $I_S(t)$ estimations by varying the slow time constant with
$\Delta \tau_S = \pm 5$ minutes has shown that $\Delta \tau_S$ only has a minor contribution to $\Delta I_S(t)$ of less than 1%, while a potential contribution
of the numerical convolution scheme itself is vanishing small. It is obvious that $\Delta I_S(t)$ is predominantly determined by the
uncertainty $\Delta S_S$ of the stoichiometry $S_S$ of the slow reaction path (Table 2)
$$\Delta I_S(t) \approx \frac{\Delta S_S(t)}{S_S(t)} \cdot I_S(t) \qquad (21)$$

The impact of the slow time constant $\tau_S$ on the stoichiometry $S_S$ and its uncertainty $\Delta S_S$ is also insignificant, as we assessed
by varying with $\Delta \tau_S = \pm 5$ minutes. Further, any contribution of the numerical schemes of deconvolution and its additional
smoothing to the uncertainty of $I_F$ have been checked and appeared to be vanishingly small (< 0.5%).

*Uncertainty contribution $\Delta \eta_C$:*
The conversion efficiency $\eta_C(P)$ (Eq. 18) has been calibrated to the OPM such that its uncertainty $\Delta \eta_C(P)$ includes also the
uncertainty of the $P_{O3,OPM}$ measurement by the OPM as follows
$$\frac{\Delta \eta_C(P)}{\eta_C(P)} = \sqrt{\frac{(\Delta a)^2 + (\text{Log}_{10}(P) \cdot \Delta b)^2}{(\eta_C(P))^2} + \left(\frac{\Delta P_{O3,OPM}(P)}{P_{O3,OPM}(P)}\right)^2} \qquad (22)$$

Hereby $\frac{\Delta P_{O3,OPM}(P)}{P_{O3,OPM}(P)}$ is the relative uncertainty of the $P_{O3,OPM}$ measurement of the OPM which is estimated to be better that 2
% at $P > 10$ hPa, and with lower pressures slightly increasing to 3 % until $P = 5$ hPa through potential small wall losses at
these pressures. The reported relative uncertainty values here for the OPM are about 1.5 % better than the values mentioned
before by Proffitt et al. (1983) because of the seven times smaller uncertainty of the new UV-absorption cross-section
(Hodges et al., 2019) compared to the former cross-section (Hearn et al., 1961) that was used before to derive the $P_{O3}$
measurement of the OPM.



The overall uncertainty budget for the TRC method is summarized in Table 5. Figure 11 shows the contributions of the
different uncertainty sources to the uncertainty budgets for the SPC6A/SST1.0 and EN-SCI/SST0.5 when applying the TRC
method for a typical mid-latitude and tropical ozone profile as used in JOSIE 2009/2010 and JOSIE 2017, respectively.  The
results for SPC6A/SST0.5 and EN-SCI/SST1.0 for JOSIE 2009/2010 and the low buffered SPC6A/SST0.1 and EN-
SCI/SST0.1 for JOSIE 2017 are shown in Figure S10 in Supplementary Material.  For the sake of clarity, the uncertainty
contributions due to (i) ascent rate variation, (ii) pressure uncertainty, (iii) total ozone normalization factor are not included
here, as these are beyond the scope of this study.  However, the characteristics of these uncertainty contributions, as reported
by Tarasick et al. (2021) and GAW Report No. 268, would not change the uncertainty budget of the TRC method itself.

**Table 5.** Sources of ozonesonde profile uncertainty and their estimated magnitudes for the TRC method. All quoted
uncertainties are one standard deviation ($1\sigma$). (*) To approximate $\Delta S_S$ as a one standard deviation uncertainty the MAD
values (only covering 25-75 percentiles) in Table 2 have been multiplied by 1.5 to become compatible with the Gaussian
error propagation applied here.

| Source | Uncertainty | Reference |
|---|---|---|
| Pump flow rate $\Phi_{P0}$ | $\Phi_{P0}$ [E-3-3] and $\Delta\Phi_{P0}$ [E-3-9] | GAW Report No. 268 (2021) |
| Pump temperature $T_P$ | $T_P$ | GAW Report No. 268 (2021) |
| Pump efficiency $\eta_P$ (P) | $\eta_P(P)$ and $\Delta\eta_P(P)$ in Table 1: JMA-efficiency | Nakano and Morofuji (2023) |
| Absorption efficiency $\eta_A$ | $\eta_A = 1.00$ and $\Delta\eta_A = 0.01$ | GAW Report No. 268 (2021) |
| Measured cell current $I_M(t)$ | $\Delta I_M(t) = \pm\ 0.005$ µA.      at $I_M(t) < 1.00$µA<br>$\Delta I_M(t) = \pm\ 0.5\%$ of $I_M(t)$  at $I_M(t) > 1.00$µA | GAW Report No. 268 (2021) |
| Background current $I_{B0}$ | $\Delta I_{B0} = 0.01$ µA | GAW Report No. 268 (2021) |
| Slow cell current $I_S(t)$ | Different sonde Type and SST:<br>$\Delta I_S(t) = \frac{\Delta S_S(t)}{S_S(t)} \cdot I_S(t)$  from Eq. (21)<br>$S_S$ and $\Delta S_S$ from Table 2 (*) | This study |
| Fast cell current $I_F(t)$ | $I_F(t)$ from Eq. (7) and $\frac{\Delta I_F}{I_F}$  from Eq. (20) | This study |
| Conversion efficiency $\eta_C$ (P) | Different sonde type and SST:<br>$\eta_C$ (P) from Table 3  and  $\frac{\Delta\eta_C(P)}{\eta_C(P)}$ from Eq. (22) | This study |
| Partial pressure ozone by OPM: $P_{O3,\ OPM}$ | $\Delta P_{O3,\ OPM}$ :  2 % at $P > 10$ hPa<br>            2 % to 3 % at $P$ from 10 hPa to 5 hPa | This study |



In both the mid-latitude and tropical case (Fig. 11) it is seen that the ("background") current in the troposphere and the
conversion efficiency in the stratosphere are the dominant uncertainty sources. For the conventional method the conversion
efficiency is based on the assumption that the overall stoichiometry factor is 1.00 with an uncertainty of 0.03 (Dietz et al.
1973), and obviously also the dominant uncertainty source in the stratosphere. However, in this study we have shown that
the overall stoichiometry can significantly differ from unity, which makes the overall uncertainty for the conventional
method rather optimistic. For the TRC-method $\Delta\eta_C(P)$ is mostly determined by the 2-3% uncertainty of the OPM as the
reference to obtain the $\eta_C(P)$ calibration functions (Table 4). In the troposphere, the contribution of $I_S$ correction in the TRC
method is mostly smaller than the $I_{B1}$ correction in the conventional method, particularly in the tropics.




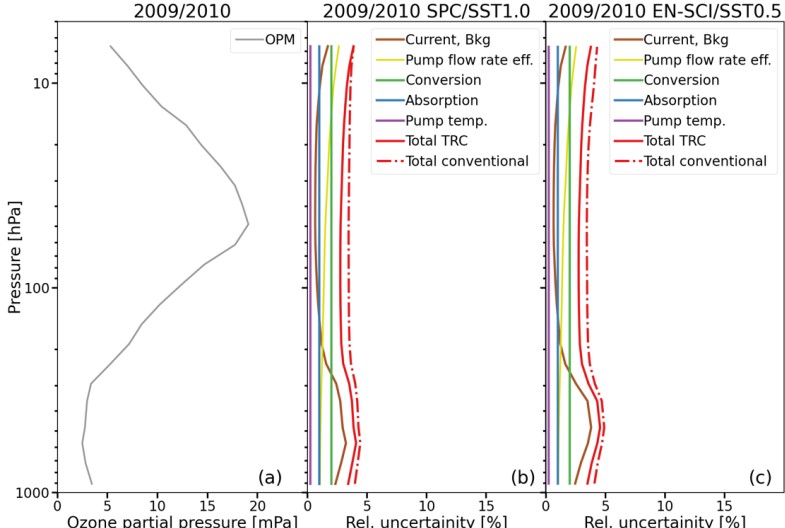


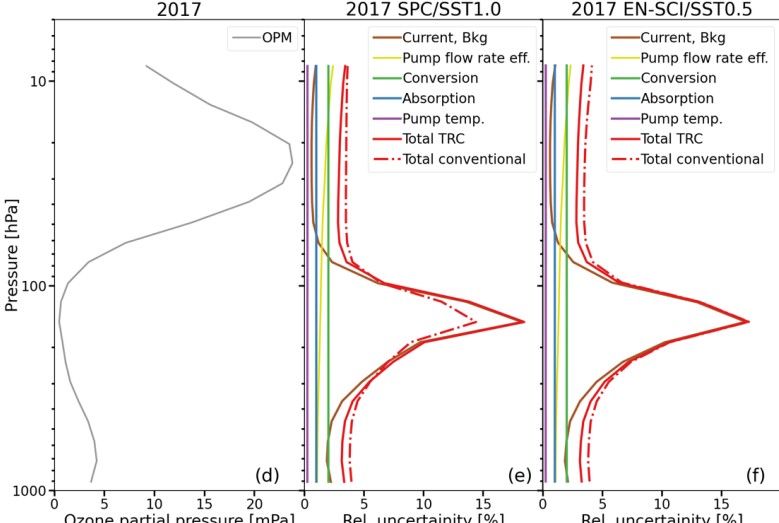


**Figure 11.** Uncertainty budgets of a mid-latitude (diagrams a, b, c: JOSIE 2009/2010) and tropical (diagrams d, e, f: JOSIE
2017) ozonesonde profile, showing the influence of the different uncertainty source terms listed in Table 5 for the TRC
method for SPC/SST1.0 (diagrams b and e) and EN-SCI/SST0.5 (diagrams c and f). Total uncertainty (red solid line)
consists of (i) Corrected cell current (brown line: $I_M$-$I_{B0}$-$I_S$ (TRC) ); (ii) Pump flow (yellow/green line: flow rate +
efficiency); (iii) Conversion efficiency (green line); (iv) Absorption efficiency (blue line); (v) Pump temperature (purple
line). In addition, total uncertainty conventional method (dashed red line).





However, both their contributions to the uncertainty are of the order of 0.01-0.02 µA, but on a relative scale they become
strongly dependent on the magnitude of the ozone partial pressures, particularly in the upper tropical troposphere. In the
stratosphere the contributions of the different uncertainties do not vary much and the overall uncertainty stays well below our
5% target.
It is to be noted that in the Tropics in the upper troposphere the partial pressure of ozone $P_{O3}$ can be in remote air conditions
very low of the order of 0.1-0.3 mPa while the detection limit of the ECC-sensor is of the order of 0.01-0.02 µA , which
corresponds to ozone levels of about 0.04-0.08 mPa. It is obvious that at these very low ozone levels the ECC-sonde
performance is strongly restricted to its detection limit, which of course can have a significant and large impact on the
overall uncertainty of the $P_{O3}$ ozonesonde measurements.

**7. Implementation of the New Time Responses Correction Into Field Operation**
A detailed procedure for applying the TRC method in practice is described in Appendix C. In this section, we apply the new
methodology developed in the previous sections to ozonesonde profile data from three different stations: (i) a mid-latitude
site (Uccle); (ii) a tropical station (American Samoa), and (iii) an ozone hole profile from the South Pole station in the
Antarctic. At those sites, we selected ascent and the corresponding descent profiles, such that the new methodology to
resolve time response effects in the ECC signal can be assessed by comparing the ascent and descent profile of the same
flight.
For the ozonesonde profiles of the three stations, we first determined the slow component $I_S(t)$ by convolution of the
measured cell current $I_M(t)$ with an exponential decay with a time constant $\tau_S$ = 25 minutes (Eq. 10) and conversion
efficiencies $S_S$ = 0.018 for SST0.5 (Uccle) and $S_S$ = 0.023 for SST0.1 (Samoa & South Pole). For the $I_S$ at time t = 0 of the
launch, (i) zero is used at Uccle, as the last exposure to ozone usually occurs at least one hour prior to launch and the
measured value will fall back to $I_{B0}$, and (ii) we use $I_{B1}$-$I_{B0}$ multiplied by the exponential decay factor $X_S$=Exp[-$\Delta t/\tau_s$],for the
other two stations, with $\tau_s$ = 25 min and $\Delta t$ = 30 min (South Pole) and 90 min (Samoa). Those time intervals are the typical
time differences between the $I_{B1}$ measurement and launch time at those sites. This slow component is then subtracted from
the measured cell current $I_M$, together with the background current $I_{B0}$. The remaining signal is the fast component, which is
deconvolved to correct for the fast time response $\tau_F$. For this latter, the time lag measurements before launch at the stations
(e.g. time to drop from 4 to 1.5 µA) are taken. The smoothing of $I_{F,D}$ is done by applying a Gaussian filter prior to the time
lag correction using a width equal to 20% of the fast time lag constant (as in Vömel et al., 2020). The final currents are then
converted to ozone partial pressures using the calibration functions in Table 4 as conversion efficiency, taking the Nakano
and Morofuji (2023) pump efficiency correction factors into account, correcting the pump temperature and the pump flow
rates as in GAW#268 (2021). For the conventional method, the GAW recommendations have been followed rigorously,
instead of subtracting $I_{B0}$ (Uccle) and $I_{B2}$ (Samoa and South Pole) as background currents.

In Fig. 12, the profiles corrected with the conventional method are on the left side, while the implementation of the "new"
TRC + calibration function on the profiles is shown on the right side. It should immediately strike the eye that the agreement
between the ascent and descent profiles is much improved after applying in particular the fast time response deconvolution
with the new method, and this for the three different sites. But also the profile shape, e.g. around the ozone peak maximum at
the Uccle and Samoa profiles, corresponds much better with each other for the ascent and descent profiles for the new
method. The slow time response correction contributes to a certain extent as well to this better profile shape agreement.






**Figure 12.** Comparison of vertical ozone profiles obtained during ascent (green solid line) and descent (purple solid line) at
three different ozonesounding stations (Uccle, Samoa, and South Pole) by applying once the conventional method (left
diagrams) and the TRC method (right diagrams).





A nice illustration of the impact of the slow time response correction is also found in the upper troposphere of the Samoa
ozone profile. The upper tropospheric ozone concentrations are significantly decreased in both the ascent and descent
profiles after applying this correction, while still agreeing very well. The strong reduction of upper-tropospheric ozone
concentrations can be ascribed to correct for previous exposure to relatively high ozone amounts from the lower troposphere
plus the (artificial) ozone spike for the ascent profile and from the ozone maximum for the descent profile.

The TRC figures are remarkable in amplifying the features after correcting for the fast time constant. We already noted that
this new method is able to resolve some features in the ozonesonde data that were effectively present in the (faster) OPM
ozone measurements in the JOSIE simulations. As noted by Vömel et al. (2020), the noise amplitude of the fast response
time-lag-corrected data is comparable to that of the original data, but its spectral characteristics are different as a result of the
smoothing algorithm. As a result, individual data points are heavily influenced by the noise characteristics of the smoothed
data. This is demonstrated by the ozone spike in the Samoa ascent, which has a larger peak amplitude for the new method.

**8.  Summary and Conclusions**
The ECC ozonesonde, in principle an absolute measuring device, encounters in the course of its flight several imperfections,
e.g. changing pump and conversion efficiency, that need to be corrected for. In the actual processing chain, the used "pump
efficiency" tables (Komhyr 1986, Komhyr et al., 1995) in fact represent an overall correction, empirically tweaked to
coincident total ozone measurements, that includes both a measured pump flow efficiency and an estimate of the
stoichiometry increase over the flight (GAW Report No.268, 2021). However, the availability of recent measured ECC pump
flow efficiencies (Nakano & Morofuji, 2023), confirming earlier measurements, together with the knowledge that the ECC
sonde response (chemical reactions pathways) is driven by a slow and fast component (Vömel et al., 2020, Tarasick et al.,
2021), call for a new approach. Vömel et al. (2020) also questioned the term "background current" in the ECC processing.

This study describes the concepts and the development of an updated methodology of ECC sonde data processing that
applies a better correction of the ozone exposure dependent stoichiometry of the $O_3$+KI titration reaction in the
electrochemical cell of the ECC-sonde through the use of true pump efficiencies combined with resolving the time responses
of the slow ($\cong 25$ min) and fast ($\cong 20$-25 sec) components of the measured ECC-ozone sensor current. Experimental evidence
is given to treat the measured ECC-sensor current as the superposition of a (i) dominant fast ozone current $I_F$ ; (ii) slow time-
variant, past ozone-exposure dependent, current $I_S$ ; (iii) a constant ozone-independent background current $I_{B0}$ .

The Time Responses Correction (TRC) method developed here is briefly described in three steps:

I.    The slow cell current component as a function of flight time is determined from the measured ozone sensor current,

after correction for the constant background current $I_{B0}$, by using a first order numerical convolution scheme (Eq.

(10). Hereby, the in-flight time response tests of JOSIE 2009/2010 have been used to quantify the stoichiometry

($O_3/I_2$) factors $S_S$ (and their uncertainties) of the slow reaction pathways for both sonde types, SPC and EN-SCI, and

two different sensing solution types, SST0.5 and SST1.0. In separate laboratory upward and downward response

time experiments $S_S$ and $\Delta S_S$ of the low buffered combination of EN-SCI with SST0.1 have been determined using

exactly the same approach as in JOSIE 2009/2010.

II.    By subtracting the constant background current before exposure of ozone ($I_{B0}$) and the time variant slow sensor

current $I_S$ from the measured ECC-sensor current $I_M$, the remaining fast sensor current $I_F$ has been resolved from the



20-30 s. time response by using a first order deconvolution scheme (Eq. 12). Essential thereby is that the resulting

deconvolved fast current $I_{F,D}$ is smoothed adequately to eliminate the high frequency noise into $I_{F,D,S}$.

III. From $I_{F,D,S}$ and using the correct pump efficiency (Table 1: Nakano and Morofuji, 2023) the partial pressure of

ozone measured by the ECC-sonde is determined (Eq. 6). Additionally, using the conversion efficiency table 4

("calibration functions"), the ozonesonde measurement is referred to the reference of the ozonesonde network, i.e.

the photometer in the simulation chamber of the WCCOS in Jülich


Because the numerical convolution scheme used here is a recursive expression, the initial condition of $I_S$ at the launch carries
the past exposure of the pre-launch preparations. In laboratory experiments it was shown that after $I_{B1}$ has been recorded
during the pre-flight preparation and the ECC pump is not running anymore, $I_S$ will further decay exponentially at the slow
time constant $\tau_S$=25 min. By knowing the time span between recording of $I_{B1}$ and turning-on the pump just before launch $I_{B1}$
can be used to derive the initial value of $I_S$ at the launch. Therefore, it is essential that during the pre-flight preparations both
background currents before ($I_{B0}$) and after ($I_{B1}$) exposure of ozone are being recorded, including the timestamp at recording
$I_{B1}$ and activating the pump just before launch of the sonde. Similarly, our understanding of this slow time constant justifies
the use of limiting values for $I_{B0}$ and after $I_{B1}$ in the operational preparation of ozone soundings, with filters providing a good
quality zero ozone air source.

The slow stoichiometry factor $S_S$ of the slow conversion of $O_3$ into $I_2$ and their MAD-uncertainties (Table 2) are each based
on a statistically relevant number of samples. $S_S$ depends on the different SSTs used (Table 2), but is not dependent on the
sonde type, which indicates that the secondary reaction pathway is not responsible for the systematic 4-5 % relative
differences existing between EN-SCI and SPC when operating with the same SST. However, a direct quantitative relation of
the buffer strength and the magnitude of $S_S$ only holds for the full buffered SST1.0 ($S_S \cong$ 0.046-0.050) and the half-buffered
SST0.5 ($S_S \cong$ 0.017-0.018), but not for the low-1/10$^{th}$ buffered SST0.1 ($S_S \cong$ 0.023). For SST0.1 significant lower $S_S$ values
would be expected, which might indicate that, in lower buffered sensing solutions, another competing chemical reaction
scheme may occur that produce also free iodine at a 25 minutes time scale and contributes to $I_S$. This may be the reason that
for non-buffered or low-buffered sensing solutions $I_{B1}$ values of 0.01-0.04 µA are still recorded.

$S_S$ values reported in Table 2 are significantly smaller than the so-called "steady bias factor" values applied by Vömel et al.
(2020), which are the overall excess stoichiometry derived from steady state experiments under ozone exposure (Vömel and
Diaz, 2010). The difference may be explained by the overall excess stoichiometry originating from the secondary reaction
pathway is only partly contributing to the slow $I_S$ and the other part is still contributing to the fast $I_F$ (Appendix A). Further,
in contrast to this study, Vömel et al. (2020) do not correct for $I_{B0}$ before determining $I_S$ and calculating $I_F$. These two
different approaches in the methodology (e.g. $I_{B0}$ subtraction and different stoichiometry factors $S_s$ for the slow current $I_S$ )
will of course lead to different results when comparing the sondes to the OPM. To demonstrate the impact of these different
assumptions between both correction schemes we have processed the JOSIE 2009/2010 and JOSIE 2017 according the TRC-
scheme used by Vömel et al. (2020). The comparisons are shown in the supplementary material in the figures S4 and S6 for
JOSIE 2009/2010 and JOSIE 2017, respectively. The impact of subtracting $I_{B0}$ is generally small and only of significance in
the upper troposphere in the Tropics, where including the $I_{B0}$ subtracting leads to better agreement with the OPM. The
impact of larger $S_S$ values for SST1.0 and SST0.5 will lower the differences to the OPM above 100 hPa, but there still
remains a significant deviation from the OPM. In the upper troposphere, the larger $S_S$ gives negative deviations, particularly
in the Tropics.



Different JOSIE data sets (JOSIE 2009/2010, JOSIE 2017, and JOSIE 1996 + 1998 + 2000 + 2002) have been used to
compare the relative differences of the sonde to the OPM obtained with the Time Responses Correction (TRC) versus the
conventional methodology of post flight data processing (GAW Reports No. 201 and 268). Hereby, it is very important to
mention that, in contrast to the conventional methodology, the relative differences obtained with TRC are almost
independent of the past ozone exposure and increases only a few percent with altitude (or lower pressure). This is most
pronounced in the tropical ozone profiles at 200-100 hPa pressure in the upper troposphere with very low ozone values and
the steep vertical ozone gradient when entering into the lower stratosphere. The typical systematic relative differences of 3-
5% for the same sonde type but different SST1.0 or SST0.5 as observed since JOSIE 2000 are still preserved in the TRC.

The different behavior between JOSIE2009/2010 and JOSIE2017 in the relative differences of the TRC corrected sonde
profiles with the OPM for pressures smaller than about 13 hPa can be ascribed to different pump temperatures used for the
mid-latitude and tropical profiles in the resp. campaigns. During JOSIE2009/2010, the higher pump temperatures led to a
higher boiling rate in this pressure range, which has been observed by the higher solution weight losses.
The TRC mean relative differences of the sonde with the OPM show a strong consistency for the different pairs of sonde
type and SST and can be therefore represented by a linear regression as function of $Log_{10}$ of the pressure. This linear
regression can be interpreted as the calibration function of the correctional term of the conversion efficiency when deviating
from one (Eq. 18). The calibration functions introduced here for the various sonde-SST combinations, parameterized as a
function of ambient air pressure in Table 4, are independent of the ozone exposure, and thus invariant to the measured ozone
profile itself. The use of these calibration functions will allow us to get the global ozonesonde records traceable to one
common standard, i.e. the OPM of the WCCOS. The origin of these calibration functions remain speculative, but there are
some experimental indications that they are linked to the unknown stoichiometry of the fast chemical conversion of $O_3$ into $I_2$
and not caused by an underestimation of the slow cell current $I_S$.

The overall uncertainty of combining the TRC with the use of the calibration functions is about 3-4 % throughout the entire
ozone profile, except for the upper troposphere, where the overall uncertainty can increase up to 10% for very low ozone
amounts, particularly in the tropics. The major uncertainty sources in the upper troposphere are the constant background
current $I_{B0}$ and the slow current $I_S$ (i.e. $S_S$), despite the correction of the slow current for the past ozone exposure in the TRC.

The TRC have been tested in practice (practical guidelines in Appendix C) for three different vertical ozone profiles
measured during ascent and descent at a mid-latitude site, a tropical station and during an ozone hole at the South Pole. The
resolving power of the fast deconvolution numerical scheme is thereby clearly demonstrated by resolving the strong delay
shift in the descent ozone profile compared with the ascent ozone profile before and after applying the TRC. However, the
examples also clearly demonstrate the importance of careful and proper smoothing of the deconvolved ozone profile. To
apply the TRC method to the time series of an ozonesonde site, a proper determination of $I_{B0}$ and $I_{B1}$ is required. Improper
filters might increase those background currents by several orders of magnitude, compromising the subtraction by the (too
high) $I_{B0}$ value throughout the entire profile and at the beginning of the profile due to the high initial value for $I_S(t_0)$. Some
more analysis is needed to formulate alternative approaches for these cases. As stated also by ASOPOS 2.0 (GAW Report
No. 268) the use of proper gas filters to provide ozone free, dry and purified air in practice at the sounding site, is very
essential in general, but also when applying the TRC data processing.

An important outcome of this study is also that the contribution of the slow current $I_S$ is not as prominent as previously
thought because TRC demonstrates that the secondary pathway involving the buffer can also contribute to the fast
stoichiometry factor to increase the fast current $I_F$ so that the uncalibrated conversion efficiency exceeds one, which is most



likely the case for SST1.0 and SST0.5. This in contrast to SST0.1, where the slow current has most likely a different
chemical origin and not an additional contribution to $I_F$, occurs so that the fast stoichiometry (i.e. conversion efficiency) does
not exceed one and is even a few percent lower. The underlying chemical mechanisms remain speculative in some cases and
the stoichiometry of the fast $O_3$+KI chemistry cannot be quantified explicitly but only expressed implicitly in the conversion
efficiency with the introduction of calibration functions (Table 4). These calibran functions can improve the
homogenization of long term ozonesonde records of the global network, making the data traceable to one ozone standard, the
OPM at the WCCOS at Jülich (Germany). Our OPM reference values have been scaled up 1.23% compared to earlier JOSIE
publications because of the revised UV ozone absorption cross-section at 254 nm (BIPM, 2022; Hodges et al., 2019). The
latter adjustment is being introduced in the global ozone network in 2024/2025.

Some specific recommendations for further research include:

1.  Regular JOSIE-campaigns at WCCOS (Jülich, Germany) are essential to check the long-term stability of the

calibrations functions reported in this study (Table 4) and to guarantee the long term traceability of global

ozonesonde records to the OPM-standard.

2.  More research is needed to understand the slow stoichiometry $S_S$ factors in more detail, particularly for the low or

no buffered sensing solutions for which the underlying chemical processes are not understood at all. A key question

hereby is also the role of KBr in the sensing solutions. This should be in conjunction with understanding the

differences observed between the methods to derive $S_S$ from either a zero-ozone or ozone exposure time response

experiment. Dedicated laboratory experiments in the WCCOS simulation chamber can accomplish this

3.  More detailed understanding of the chemical reaction mechanisms that are responsible for the fast and slow cell

current response of the ECC-sensor, and their interaction. This should include determining the temperature

dependency of the KI+$O_3$ chemistry.

4.  Better knowledge of the time behaviour of the high background currents $I_{B0}$ and $I_{B1}$ that are often measured in

practice at the sounding sites when not using proper gas filters. Experiments are necessary to describe and

eventually correct for this high $I_{B0}$ and $I_{B1}$ caused using inadequate gas filters because re-processing ozonesonde

records often means correcting very high $I_{B0}$ and $I_{B1}$.

This study did not solve the systematic 3-5% offsets in measured ozone between EN-SCI and SPC instruments when
operating with the same SST. However, we showed that the $S_S$ values are comparable for both sondes with the same SST,
which means the differences are not caused by the slow chemistry. More research here is essential.
**Acknowledgements**
For the JOSIE 2009/2010 we are very grateful to Marcel Berg (FZJ/IEK-8, Germany) and Dr. Johannes Staufer (ETHZ,
Switzerland) for the pre-flight preparation of the ozonesondes. Many thanks to Dr. Holger Vömel for stimulating discussions
in preparing the manuscript. Also many thanks to the people who supplied ECC-sondes to be "flown" in the simulation
chamber in JOSIE 2009/2010 and 2017. For JOSIE 2009/2010 we thank: Dr. Bryan Johnson (NOAA-GML, USA), Francis
Schmidlin (NASA/Goddard/Wallops Flight Facility, USA), Dr. Hugo De Backer (RMI, Belgium), Dr. Rene Stübi (Meteo
Suisse., Switzerland), Dr. Rigel Kivi (FMI, Finland), Dr. Richard Querel (NIWA, New Zealand), Dr. Matt Tully (BOM,
Australia), Dr. Emilio Cuevas (AEMET, Spain). Sondes for JOSIE-2017 were supplied by FZJ/IEK-8, NOAA/GML and
NASA/Goddard. Researchers from FZ-J, NASA/Goddard, NOAA/GML, MeteoSwiss, RMI (Belgium), KNMI
(Netherlands), JMA (Japan), Environment & Climate Change Canada along with 8 SHADOZ operators contributed time to
JOSIE-2017. We thank the United Nations Environmental Programme, WMO, EN-SCI and SPC for supporting the



participation of the SHADOZ personnel in JOSIE-2017. JOSIE 2009/2010 and 2017 were sponsored by WMO/GAW and
Forschungszentrum Jülich (FZJ) and NASA/GSFC.

**Competing interests**

R. Van Malderen is a member of the editorial board of Atmospheric Measurement Techniques. The peer review process will
be guided by an independent editor. The authors have no other competing interests to declare.

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



| 1478 | **Appendix A: KI + O3 Chemistry in Presence of Phosphate-Buffer (NBKI after Saltzman & Gilbert, 1959)** |

Iodometric determination of ozone and the underlying oxidation of iodide ion by ozone to liberate iodine has long been
subject of controversy. The reaction of KI with $O_3$ may proceed through a variety of chemical pathways strongly depending
on pH, KI and $O_3$ concentrations, whether or not in presence of a pH-buffer. In this study the focus is on the NBKI method
and its application in the ECC-ozone sensor. Experimentally it was shown by several investigators (e.g. Saltzman and
Gilbert, 1959; Flamm and Anderson, 1975) that iodate ($IO_3^-$) as intermediate can be excluded as long as ozone partial
pressures in the air are well below 100 mPa. This makes it most likely that much of the behaviour of the ECC and its slow
and fast sensor currents may be explained by the chemical reaction mechanisms for the NBKI (Neutral Buffered KI) and its
impact of the phosphate buffer as postulated by Saltzman and Gilbert (1959). It was experimentally shown that the fast and
slow reactions increase as KI concentrations increases, whereby the slow reactions increase with the buffer concentration.
Buffered solutions with no KI show no evidence of gaseous $O_3$ uptake into the sensing solution, indicating that the additional
reactions with $O_3$ are secondary reactions after the initial $O_3$ + KI reaction.

Primary reaction pathway:
(R1)   $2KI$   $+ H_2O$   $+ O_3$   $\rightarrow 2KOH + I_2 + O_2$
*In ion-notation:*
(R2)   $O_3$   $+ 2H^+$   $+ 2I^-$   $\rightarrow O_2 + I_2 + H_2O$
Or in detail (postulated after Saltzman & Gilbert, 1959) :
(R3)   $O_3$   $+ I^-$   $\rightarrow IO^- + O_2^*$   (fast)
(R4)   $IO^-$   $+ I^- + 2H^+$   $\rightarrow I_2$   $+ H_2O$   (fast, neutral/acid)
(R5)   $O_2^*$   $+ M$   $\rightarrow O_2$   $+ M$   (fast)
Losses of $IO^-$, i.e. $I_2$:
(R6)   $IO^-$   $+ IO^-$   $\rightarrow 2I^-$   $+ O_2$   (slow)

• If all $O_3$ would be absorbed and react with KI in this primary reaction pathway, it would be expected that the
stoichiometry for $O_3/IO^-$ i.e. $O_3/I_2$ in neutral/acid solution is equal to one.
• However, self-reaction of $IO^-$ (R6) can be a loss mechanism, competing with the formation of $I_2$ (R4).
• In general, loss mechanisms of $IO^-$ might compete with (R4) and then the stoichiometry of primary reaction pathway is
less than one.
• ECC shows for 1% KI and no buffer a stoichiometry less than one (Johnson et al., JGR, 2002).
• Dismutation (disproportioning) of $IO^-$ into iodate ($IO_3^-$) and $I^-$ is extremely slow and is of no importance in case of the
ECC-sensor. Iodate-chemistry plays first a role at significant higher KI or $O_3$ concentrations than are used in the ECC-
sensor or encountered in the atmosphere, respectively.

Secondary Reaction Pathway: Impact of Phosphate Buffer
(R7)   $O_2^*$   $+ I^-$   $+ H_2PO_4^-$   $\rightarrow IO^-$   $+ H_2PO_5^-$   (fast)
(R8)   $H_2PO_5^-$   $+ I^-$   $\rightarrow H_2PO_4^-$   $+ IO^-$   (slow)
(R4)   $IO^-$   $+ I^-$   $+ 2H^+$   $\rightarrow I_2$   $+ H_2O.$   (fast)
But also losses of $I_2$ iodine (via $IO^-$ losses):
(R9)   $H_2PO_5^-$   $+ IO^-$   $\rightarrow H_2PO_4^-$   $+ I^-$   $+ O_2$   (slow)
(R6)   $IO^-$   $+ IO^-$   $\rightarrow 2I^-$   $+ O_2.$   (slow)
• R7 is the key reaction to form extra $IO^-$ that can react via (R4) into $I_2$ and is contributing in addition to the fast reaction
pathway and thus adding to the stoichiometry causing the fast ECC signal.





•  $H_2PO_5^-$ can be seen as the interim reactant that is formed fast but via (R8) decaying slowly to form extra $IO^-$. This latter

can produce in addition extra $I_2$ which is causing the slow part of the ECC current.

•  It is known that $H_2PO_5^-$ reacts similar as $H_2O_2$ to form $IO^-$, i.e. $I_2$ with typical time constant of about 25 minutes: this fits

to the slow, secondary response time of ECC of typical 25 minutes.

**Appendix B: Laboratory Experiments to Determine $S_s$ for EN-SCI SST0.1**
As no time response tests are available during JOSIE campaigns for SST0.1 to determine $S_S$, we undertook laboratory
measurements under room conditions in Uccle (Belgium). During the experiments, 4 ozonesondes were simultaneously
exposed to ozone amounts generated by a photometric ozone calibrator Teledyne API T703 according to the following
scheme (3 times): 30 minutes of exposure to a value of 450 µg/m³ (around 225 ppb) ozone were preceded and succeeded by
10 minutes of ozone-free air, see Fig. B1. The value of 450 µg/m³ has been imposed by the upper limit (6.5 µA) of the
microcurrent meters used in the Forschungszentrum Jülich homemade ground calibration box for the 4 ozonesondes. These
microcurrents were read out digitally and, as in the JOSIE experiments, the $S_S$ values were again estimated as the average
over a 50s time interval between 4 and 5 minutes after the end of the ozone exposure. As the time response test intervals in
these laboratory measurements are twice as long (10 minutes) as in the JOSIE 2009/2010 campaigns, we tried different
timings for the determination of the $S_S$ values, but they did not give significantly different results for the slow stoichiometry
coefficients. Again, the differences between the $S_S$ values obtained from the different time response test intervals in one
experiment were insignificant as well.

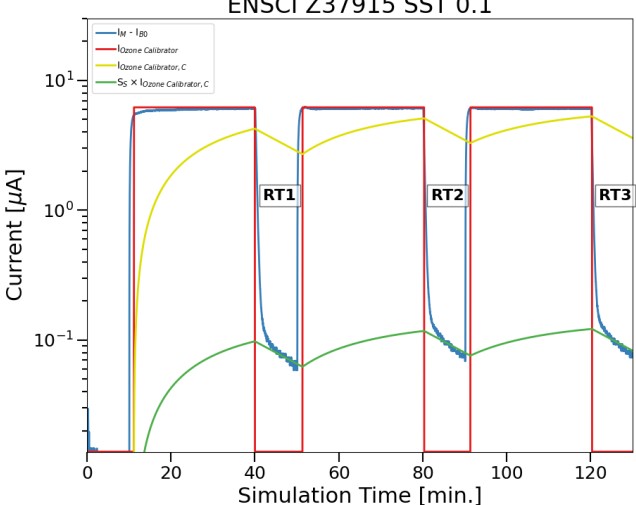


**Figure B1.** Example of a series of three upward and downward ozone steps generated by a photometric ozone calibrator
Teledyne API T70 (represented by the generic $I_{Ozone\ Calibrator}$: red line) and the response of the measured cell current $I_M$-$I_{B0}$
(blue line) of an EN-SCI SST01 ozonesonde as function of time, the 25 min convolved $I_{Ozone\ Calibrator,\ C}$ (yellow line) and the
slow current after determination and application of $S_S$ ($S_S$ x $I_{Ozone\ Calibrator,\ C}$ : green line).

In total, we have 8 $S_S$ estimations with 4 EN-SCI ozonesondes filled with SST0.1 solutions coming from 3 different
experiment runs: 2 runs with each 2 (new) EN-SCI ozonesondes (with SST0.1), and a run with all 4 (re-used) EN-SCI
ozonesondes involved. These 4 ozonesondes, all with serial numbers Z379xxx, have been prepared by the same person,
according to the SOPs defined in GAW Report No.268, 2021. The median value for $S_S$ for the 8 experiments, each including





three time intervals, is 0.023 ± 0.005. This value is very close to the value $S_S$=0.017 found for SST0.5 during the JOSIE
2009/2010 campaign, whereas a smaller value could be expected due to the lower buffer amount in SST0.1 (see Johnson et
al., 2002 and Sect. 3.2). However, the same Uccle experimental setup and method as described here above for EN-SCI
SST0.1 have been used to determine the $S_S$ coefficient for 4 EN-SCI ozonesondes filled with SST0.5 (serial numbers
Z379xxx, but different from those used with SST0.1) during two experimental runs. The resulting median value,
0.022±0.004, is again in close agreement with the value determined for EN-SCI SST0.5 with the JOSIE 2009/2010 (0.018 ±
0.004), confirming the consistency between the two instrumental setups to determine the stoichiometry coefficients.
Nevertheless, a JOSIE campaign is foreseen in 2023 to determine the $S_S$ factors for SST0.1 for both EN-SCI and SPC
ozonesondes, using the same simulation setup as in JOSIE 2009/2010.
**Appendix C: How to use TRC + calibration functions in practice: Practical Guidelines**
In this appendix, we give a schematic overview of the different steps that need to be taken to implement the TRC +
calibration functions in the data processing of an ozonesonde time series in practice, displayed schematically in the flow
chart in Fig. C1.

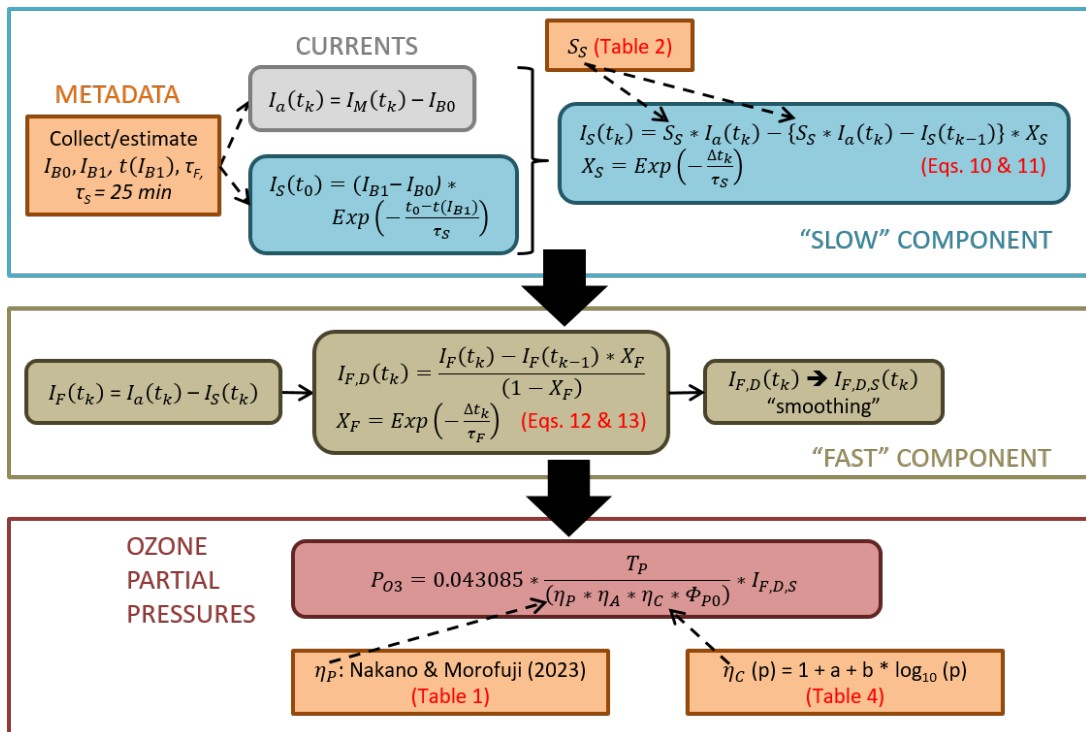


**Figure C1.** Flow chart summarizing the processing steps for the Time Responses Resolving Method (TRC) for correcting
ozonesonde data. The table and equation numbers in red refer to these in this paper.

First, it should be noted that the TRC is applied on the currents measured by the ozonesonde. Hence, these ozonesonde's raw
measurements should be available. Normally, when a site has been homogenized as part of the O3S-DQA activity, the
currents have been made available or have been converted back from the ozone partial pressures. Secondly, the TRC



demands the knowledge of some metadata parameters that should have been measured during the preparation of the
ozonesonde 0-1 day prior to launch (see also Fig. C1): $I_{B0}$, $I_{B1}$, the time of the $I_{B1}$ measurement (relative to the launch time),
and the sensor fast response time $\tau_F$, measured as the time to drop from 4.0 to 1.5 µA (after the 5 µA test). If those metadata
parameters are missing, these might be estimated as the means over a representative time period, e.g. using the same filter for
determining the background currents, or the same batch of ozonesonde serial numbers or sensing solution for the fast
response time.
In a next step, the $I_{B0}$ value is subtracted from the time series of measured currents of the sounding, resulting in $I_a(t_k)$, and all
forthcoming calculations should be done with those currents $I_a(t_k)$. As the calculation for obtaining the slow component of
the ECC signal is a recursive equation (Eq. 10), the slow component at launch time should be estimated first. Therefore, it
suffices to start from the last measured value of the ozonesonde before launch, the $I_{B1}$, corrected for (i.e. subtract) the $I_{B0}$
value, and convolve it with an exponential decay function with a slow time constant of 25 minutes. Hereby, the time
difference between the $I_{B1}$ measurement and the launch is used. If this time difference is large enough (GAW Report No. 268
) recommends a minimum 30-min time window), the exponential decay function will be close to zero, $I_{B1}$ will approach the
$I_{B0}$ value, and the slow component at launch time will be zero, which is the allowed lower limit.  Now, for every time step,
the slow component of the ECC signal can be calculated from equations 10 and 11, using the stoichiometry factor $S_S$ from
the sonde–SST combination (see Table 2). This slow component can be seen as a time varying background current and
should be subtracted from the currents $I_a(t_k)$, to be left over with the fast component $I_F$ of the ECC signal.
To eliminate the 20 to 25 seconds response delay in the fast component, the latter can be deconvolved (Eqs. 12 and 13), i.e.
corrected for the exponential decay of the signal with the fast sensor response time, measured before launch. This
deconvolution will introduce a lot of noise in the signal, and therefore, a smoothing of the current, either before or after the
deconvolution, will be necessary. Different smoothing algorithms can be considered, with different filter widths and/or time
windows (e.g. for running averages). The choice of the smoothing algorithm depends on the application, e.g., to resolve
steep vertical gradients and the profiles (smooth mid-latitude vs. upper-tropospheric tropical profile), as well as the
measurement time interval (10 s versus 1s time resolution). At the end, a compromise between the smoothness of the profile
and a full correction for the time response delay around strong vertical gradients should be sought.
The smoothed, deconvolved time series of the fast component $I_{F,D,S}$ of the ECC signal is then used in the basic equation of
the ozonesonde signal, converting the current to ozone partial pressure. In this equation, the recommended corrections for $T_P$,
$\eta_A$, and $\phi_{P0}$ in  GAW Report No. 268 should be implemented as well: the conversion to the piston pump temperature [E-3-
15], a correction for the absorption efficiency if the cathode cell was only filled with 2.5 cm$^3$ of solution before launch [E-3-
11-A&B], and the humidification [E-3-4] and pump temperature [E-3-7] corrections for the pump flow rate at the ground. In
comparison with the recommended processing in GAW Report No. 268, the pump efficiency corrections proposed by
Nakano & Morofuji (2023) should now be used for all combinations of sonde type and SSTs, as these are the actual
measured ones. The Komhyr (1986) and Komhyr et al. (1995) tables should be discarded, as these are empirical correction
curves, as they actually combine pump efficiency and conversion efficiency. A last difference with the conventional method
as proposed in GAW Report No. 268 is the use of the "calibration functions" defined in Sect. 6, Eq. 18: $\eta_C(p) = 1 + a + b *$
$log_{10}(p)$, with the coefficients $a$ and $b$ determined for every sonde type and SST combination separately (see Table 4), for the
conversion efficiency, instead of adopting the value $\eta_C(p) = 1.00$. Using the calibration functions, the ozone sounding
measurement should be traceable to the common reference of the ozonesonde network, the ozone photometer OPM in the
simulation chamber of the World Calibration Centre for Ozonesondes in Jülich.

To calculate the uncertainties associated with the ozone partial pressure measurements of an ozonesonde, corrected with
TRC + calibration functions, the uncertainty equation E-3-1 in GAW Report No. 268 (2021) forms the basis. With respect to



this formula, the uncertainty equation for the TRC (see also Fig. C2) has one changed term, and the meaning of a couple
other terms has changed. We will only describe these 3 terms here.

**Figure C2.** Overview of the different data processing steps and input to derive the uncertainty of the ozone partial pressure
measured with an ozonesonde, using the TRC + calibration functions. Figure adapted from Fig. C-4 in GAW Report No. 268
(2021). The equation numbers also refer to equations in this GAW report. Table numbers in red refer to tables in the main
text of this paper.
First, as both the $I_{B0}$ and slow component $I_S$ are subtracted from the measurement background in the TRC, the uncertainties
of the $I_{B0}$ and $I_S$ should be included now. For $I_{B0}$, the uncertainty is estimated to be 0.01 µA, and the (relative) uncertainty of
the slow component is, in a first order approximation, equal to the (relative) uncertainty of the stoichiometry coefficient $S_S$.
The uncertainties of $S_S$ for the different SSTs can be found in Table 2.
For TRC, the uncertainty of the pump efficiencies $\Delta\eta_P$ are now equal to the standard deviations of the pump efficiency
measurements reported in Nakano & Morofuji (2023), also shown in Table 1. Finally, the uncertainty of the conversion
efficiency is no longer estimated as a fixed value $\Delta\eta_C = 0.03$, but should take into account the uncertainty of the derived
calibration functions $\eta_C(p) = 1 + a + b * log_{10}(p)$ in Sect. 6 (see Table 4 for the uncertainties on the linear regression
coefficients a and b for the different combinations of sonde type and SST), as well as the uncertainty of the photometer
(OPM) to which the ozonesonde measurements are traced back. This latter (relative) uncertainty $\frac{\Delta P_{O3,OPM}(P)}{P_{O3,OPM}(P)}$ is estimated to
be around 2%.



**Appendix D: Nomenclature of parameters**
$I_{B0}$      Background Current before exposure with ozone (after 10 min flushing cathode cell with "zero" air)
$I_{B1}$      Background Current after exposure with ozone (after 10 min flushing cathode cell with "zero" air)
$I_{B2}$      Background Current at launch site just before flight
$I_B$      Background Current used in data processing in Eq. (1).

$S_F$      Stoichiometry factor of fast reaction pathway of conversion of $O_3$ into $I_2$
$S_S$      Stoichiometry factor of slow reaction pathway of conversion of $O_3$ into $I_2$
$I_M$      Measured (cathode) cell current
$I_{OPM}$      Ozone equivalent ECC current at time t derived from OPM
$I_F$      Fast cell current
$I_{F,D}$      Fast cell current, deconvolved
$I_{F,D,S}$      Fast cell current, deconvolved, smoothed
$I_S$      Slow cell current
$P_{O3}$      Ozone partial pressure
$R$      Universal gas constant
$F$      Faraday constant
$T_P$      Pump temperature
$\Phi_{P0}$      Pump flowrate
$\eta_A$      Absorption efficiency
$\eta_P$      Pump efficiency
$\eta_C$      Conversion efficiency
$\eta_T$      Total (overall) efficiency
$\tau_F$      Response time of fast reaction pathway of conversion of $O_3$ into fast cell current component
$\tau_S$      Response time of slow reaction pathway of conversion of $O_3$ into slow cell current component
**RT1, RT2, RT3, RT4**      Response time tests in vertical ozone profile




| 1663 | **Appendix E: List of Abbreviations (Green marked are mentioned in manuscript)** | |
|------|--------|---------------------------------------------------|
| 1664 | **ASOPOS** | Assessment of Standard Operating Procedures for OzoneSondes |
| 1665 | **BESOS** | Balloon Experiment on Standards for OzoneSondes |
| 1666 | **CMDL** | Climate Monitoring and Diagnostics Lab (formerly called GMD, now GML) |
| 1667 | **ECC** | Electrochemical Concentration Cell |
| 1668 | **EN-SCI** | Environmental Science Corporation; ECC ozonesonde manufacturer |
| 1669 | **ESRL** | Earth System Research Laboratories |
| 1670 | **FZJ** | ForschungsZentrum Jülich |
| 1671 | **GAW** | Global Atmospheric Watch |
| 1672 | **GML** | Global Monitoring Laboratory (division of NOAA's ESRL; formerly GMD) |
| 1673 | **$H_2O_2$** | Hydrogen peroxide |
| 1674 | **IAP** | Institute of Atmospheric Physics, Beijing, China |
| 1675 | **IPCC** | Intergovernmental Panel on Climate Change |
| 1676 | **JMA** | Japanese Meteorological Agency |
| 1677 | **JOSIE** | Jülich OzoneSonde Intercomparison Experiment |
| 1678 | **KI** | Potassium Iodide |
| 1679 | **NASA** | National Aeronautics and Space Administration |
| 1680 | **NBKI** | Neutral-Buffered Potassium Iodide |
| 1681 | **NDACC** | Network for the Detection of Atmospheric Composition Change |
| 1682 | **NOAA** | National Oceanic and Atmospheric Administration |
| 1683 | **$NO_x$** | Nitrogen Oxides |
| 1684 | **O3S-DQA** | OzoneSonde-Data Quality Assessment |
| 1685 | **OPM** | Ozone PhotoMeter instrument (used as UV-reference at WCCOS) |
| 1686 | **SHADOZ** | Southern Hemisphere ADditional OZonesonde |
| 1687 | **SI²N** | Ozone trend assessment study supported by SPARC, IOC, IGACO, and NDACC |
| 1688 | **SOP** | Standard Operating Procedure |
| 1689 | **SPARC** | Stratosphere-troposphere Processes And their Role in Climate |
| 1690 | **SPC** | Science Pump Corporation; ECC ozonesonde manufacturer |
| 1691 | **SST** | Sensing Solution Type |
| 1692 | **SST0.1** | 1.0% KI & 1/10th buffer solution |
| 1693 | **SST0.5** | 0.5% KI & half pH-buffer solution |
| 1694 | **SST1.0** | 1.0% KI & full pH-buffer solution |
| 1695 | **SST2.0** | 2.0% KI & non-pH-buffered solution with no KBr |
| 1696 | **STP** | Standard Temperature (=273.15 K) and Pressure (=1013.25 hPa) conditions |
| 1697 | **TOAR** | Tropospheric Ozone Assessment Report |
| 1698 | **TRC** | Time Responses Resolving Methodology |
| 1699 | **UNEP** | United Nations Environment Programme |
| 1700 | **UV** | Ultraviolet |
| 1701 | **UWYO** | University of Wyoming |
| 1702 | **VOC** | Volatile Organic Compound |
| 1703 | **WCCOS** | World Calibration Centre for OzoneSondes |
| 1704 | **WMO** | World Meteorological Organization |