# Peer review of "New Insights From The Jülich Ozone-Sonde Intercomparison 2 Experiments: Calibration Functions Traceable To One Ozone Reference 3 Instrument 4 Herman G.J. Smit1, Deniz Poyraz2, Roeland Van Malderen2, Anne M. Thompson3,4, David W. Tarasick5, Ryan M. 5 6 Stauffer3, Bryan J. Johnson6, Debra E. Kollonige3,7 7 8 1Forschungszentrum Jülich, Institute of Energy and Climate Research, IEK-8: Troposphere, Jül"

_EGUsphere, 2023_

## Author Comment (AC1)

**Review of Smit et al., *New Insights From The Jülich Ozone-Sonde Intercomparison Experiments: Calibration Functions Traceable To One Ozone Reference Instrument***

**Anonymous Referee #1, 30 July 2023**

*We thank referee#1 for her/his complete and thoughtful review of our manuscript and providing thoughtful comments and suggestions that have helped us improve this manuscript. We also thank Editor Birgit Hassler for handling our paper and coordinating the reviews. Our responses to reviewer comments are provided below in red italic text.*

**RC1**: **'Important paper on improved processing for ECC ozone sondes',**

**Overall remarks**

This is a very comprehensive, long manuscript summarizing the experiences from decades of working with ozonesondes. The authors provide a complete recipe for a better processing of ECC ozone sonde data. While not completely new, the suggested better processing now gives a reference framework for the global network of ECC ozone soundings. It also ties the sondes to the reference ozone UV photometer at the Jülich World Calibration Center.

While a bit longish in places, I find the paper quite readable. The topic certainly deserves publication and the mansucript is basically ready for publication. I only have a number of minor suggestions.

I am, however, no expert in the complex chemistry of the ECC sonde ozone measurement. Therefore it would be good to also have an opinion from someone expert in the chemistry of buffered KI solutions.

**Suggestions for improvements and clarifications**

Line 22: "cell current has not been fully quantified, resulting in time-varying background current and pump efficiencies" suggest to change to "cell current still needs to be quantified better, using time-varying background current and more appropriate pump efficiencies"

*>>> Done*

Line 24: "with the fast and the slow reactions pathways" implies that these pathways are well known / accepted. I suggest to drop both the's: "with fast and slow reactions pathways", which is much definitive and, to me, more appropriate

*>>> Done*

Line 124: change "original and homogenized data showed that significant systematic errors were eliminated" to "original and homogenized data allowed elimination of significant systematic errors"

*>>> Done*

Line 262: It would be better to say "which combine decreasing pump efficiency, increasing conversion efficiency, and typcial memory effects in the background current"

Line 280: Add "ozone" before "current"

>>> *Done*

Line 285: replace "consequently" by "at the same time".

>>> *Done*

Line 287: replace "show" by "have shown"

>>> *Done*

Line 295: replace "such" by "and"?

>>> *Done*

Line 318: replace "with the" by "to"

>>> *Done*

Line 321: add ", as observed by Johnson (2002)" after "larger than 1.0"

>>> *Done*

Figure 1b: In its current form, this Figure does not provide convincing evidence for a 25 minute decay time. The dashed red line for a 25 minute decay drops much faster than the plotted percentile lines. I think it would be much better to either subtract appropriate IB0s from the percentile lines, or to add an appropriate IB0 to the dashed 25 minute decay line.

>>> *Thanks for the suggestion. We have modified the figure by adding a new curve: IM(t)-IB0 and at t=10 min. we add the 25 min slow response (i.e. 25 min 1/e- decay) that match very close the measured IM(t)-IB0 curve. We have changed the text in the manuscript accordingly.*

Line 491 and many other places in the manuscript: It would be better to call the Komhyr "pump" corrections not "pump" corrections but something like "effective efficiency corrections", throughout the manuscript. One of the major points of the manuscript is, after all, that the Komhyr correction lump a number of things into an overall correction (which is not so far from the results of the new TRC method).

>>> *Good point and we followed your suggestion and referred the Komhyr tables now as "empirical effective K86-Efficiency and K95-Efficiency", respectively. Also in Table 1 caption and corresponding text in Line 264/265.*

Line 507: replace "vanishing" by "very small" or "negligible"

>>> *Done*

Line 509: replace "provides the measure of the true value" by "provide the true value"

>>> *Done*

Line 512, 705 and many other places in the manuscript: add "true" to "pump efficiency" when the measured efficiencies from Nakano and Morofuji (2023) or Johnson (2002) are meant,

as opposed to the "lumped" or "effective" (not-pump-only) efficiencies from Komhyr (1986 and 1995).

>>> *We followed your suggestion and added "true" when referring to the pump efficieny tables by Nakano et al. (2023) or Johnson et al (2002).*

Line 547: "independent of the KI concentration". Why? Seems to be that it could also be dependent on KI concentration, which is also halved. And for SST0.1, with 1%KI, the proportionality with buffer concentration does not hold either, and the measured values are somewhere between the values for the 1% and 0.5% KI concentration solutions. I think there is still a lot unclear here, Saltzman and Gilbert (1959) does not provide the complete answer. I suggest that the authors reword this paragraph (and similar text in other places). Here, for example, replace "independent of the KI concentration ... has been explained" by "which might be explained".

>>> *The primary cause for the change of the stoichiometry is the buffer strength and only a minor contribution by the KI-concentration. This has experimentally been shown by Johnson et al. (2002). Also, in simulation experiments done prior to JOSIE 2000, it already was demonstrated that the ECC-response by changing only the KI-concentration and having the same buffer strength is rather similar to each other. However, we have added at end of Line 547 an extra sentence to make this more clear:*

>>> *„Johnson et al. (2002) have demonstrated that an increase of the stoichiometry is primarily caused by the buffer strength with only a minor contribution by the KI-concentration. This results might be explained by …………"*

>>> *You are right that not everything can be explained in detail by Saltzman and Gilbert (1959) and certainly more research is needed, particularly in the understanding of the underlying chemical processes. We have mentioned that also in the conclusions in Chapter 8.*

Line 612,613: delete ", for example in the JOSIE simulation chamber". Seems redundandent.

>>> *Done*

Line 655: change "may decay be due" to "may be due"

>>> *Done*

Line 705: Better to say "In contrast" instead of "As opposed"

>>> *Done*

Line 714: delete "here presented"

>>> *Done*

Line 773/774: I find this description poor. Suggest to change it to "For both SPC and EN-SCI, SST0.1 sonde values are slight lower than OPM values in the stratosphere, and up to 10% lower than sonde values for the SOP recommended solutions (SPC/SST1.0 and EN-SCI/SST0.5).

*>>> Suggestion included.*

Line 808, Section 4.3: I found this confusing here. This section 4.3 and Table seems to belong much more to Section 5 Conversion efficiency, or to Section 5.1 I strongly suggest to move the text of section 4.3 and Table 3 to these more appropriate places.

*>>> Good point: we moved section 4.3 incl. table 3 into new section 5.1 and old section numbering  5.1 and 5.2 now 5.2 and 5.3 respectively.*

Table 4 and surrounding text: Of course these numbers and the conversion efficiency correction assume that the Nakano and Morofuji (2023) pump efficiency correction is perfectly correct. I think that needs to be stated.

*>>> We included at Line 865 an extra sentence stating this:*

*"Of course, the numbers are for the calibration functions, i.e. H. the conversion efficiencies, directly linked to the pump efficiency values used, and it is assumed here that the average pump efficiency values from Nakano and Morofuji (2023) in Table 4 are correct within their uncertainties and representative of this study. However, if known pump efficiency values have changed over time, the calibration functions must be adjusted accordingly."*

Section 6.1: I think the conversion efficiency correction from Table 4 is missing in this discussion and in Fig. 10. Should that not be added here?

*>>> Good point, we included the conversion efficiency correction in Fig.10 and added it in the discussion around Fig.10.*

Figure 10: I suggest to add one or two curves for the effective efficiencies from Komhyr (1986 and/or 1995), maybe as a thick curve or a shaded region in the background. (For the tropics, this may also require an "old" constant background subtraction.) At least discuss in the text, how the new effective overall corrections compare to the old Komhyr (1986 and 1995) corrections.

*>>> We included the calibrated conversion efficiency correction in Fig.10 and have add it in the discussion around Fig.10. We added for the JOSIE 2009/2010 (mid-latitude) and JOSIE 2017 (tropical) each an extra diagram as an example to show the relative corrections made for the conventional method. At the end of the section we added a paragrapgh discussing the differences of the total corrections between TRCC and the conventional method.*

Table 5: for some uncertainty sources I see values in column 2, for others I don't see values. I think it would be good to have an additional column that gives explicit values (or ranges) for each source of uncertainty.

*>>> In column 2 of Table 5 we included, where possible, for the different uncertainty sources explicit values (or ranges) for the corresponding uncertainties. For more details Figure 11 (now with log scale) serves.*

Figure 11: For the two panels on the right, it would be much better and much clearer to have a log-scale for the uncertainty. With the current linear scale it is very difficult to see values for most of the smaller uncertainties.

*>>> For the relative uncertainty we changed the linear scale into a log-scale.*

Line 1051/1052: I suggest to just say "stays well below 5%". You may have all kinds of targets, in the end the question is what can be achieved with current ECC sondes (or any other instrument).

>>> *Done*

Lines 1126 to 1132: Would be good to give some numbers for the slow current here. Something like "The slow current typically amounts to xx% to yy% of the true ozone current, but can reach 10% in regions with very low ozone."

>>> *We add following sentence: "Depending on the buffer strength the slow current typically amounts about 1-4% for SST0.5 or SST0.1 and about 2-8% for SST1.0, however, in regions with very low ozone it can reach up to 10-15 %".*

Line 1135: replace "thereby" by "for this procedure"

>>> *Done*

Line 1136: replace "is smoothed" by "has to be smoothed". I also suggest to change "the high frequency noise into IFDS." to just "high frequency noise".

>>> *Done*

Line 1152: need to add "background due to" before "conversion"? Also: I forgot what MAD is. Please spell out.

>>> *Done*

Line 1158: replace "would be" by "might be". Also: Consider my comment for line 547.

>>> *Done*

Line 1177 to line 1184: I am confused here, especially by the statement "in contrast to the conventional methodology the relative differences obtained with TRC are almost independent of the past ozone exposure". Were not the memory effects in the background one of the key aspects of the TRC method. OK now, on third reading, I get it: You are talking about differences to the OPM. So this entire paragraph seems confusing. Would it not be better to reword, and to say that JOSIE data have used to determine the free parameters of the TRC method, and now you get TRC ozone sonde data consistent with OPM values throughout all JOSIE campaigns. (With the minor exceptions mentioned in the next paragraph).

>>> *In this paragraph, it is primarily our intention to demonstrate that the observed TRC differences with OPM are independent of the ozone profile type or ozone amounts, which means that the TRC differences are independent of the past ozone exposure. This is in contrast of the conventional method, whereby the differences are dependent of the past ozone exposure. With this paragraph it is not our intention only to achieve with TRC that all JOSIE data fits to the OPM. To achieve more clarity, however, we changed the sentence started in Line 1179 and ending in Line 1181 by*

>>> *"Hereby, it is very important to mention that, in contrast to the conventional methodology, the relative differences obtained with TRC are almost independent of the ozone profile type (e.g. mid-latitude or tropical). In other words, the observed relative differences with TRC are*

*independent of the past ozone exposure and increases only a few percent with altitude (or lower pressure)."*

Line 1187: replace "can be" by "is". That is what you are doing.

*>>> Done*

Line 1189: replace "which has been observed by the" with "confirmed by"

*>>> Done*

Line 1192/93: replace "of the correctional term of the conversion efficiency when deviating from one." by "for conversion efficiency (which is not quite equal to one)".

*>>> Done*

Line 1195: replace "will allow us" by "makes"

*>>> Done*

Line 1200: delete "the use of"

*>>> Done*

Line 1203: delete "despite .... TRC". Not needed.

*>>> Done*

Line 1207: delete "thereby", replace "resolving" by "removing"

*>>> Done*

Line 1210/1211: "Improper filters", in this context this could also mean numerical filters. I would suggest to say: "Imperfect or defective zero ozone air filters"

*>>> Done*

Line 1217: replace "as prominent" by "as large". I don't think it is prominent, since it still a fairly small background in most cases.

*>>> Done*

Line 1218: after "thought" add "(Voemel, 2020)"

*>>> Done*

---

## Author Comment (AC2)

**Review of Smit et al., *New Insights From The Jülich Ozone-Sonde Intercomparison Experiments: Calibration Functions Traceable To One Ozone Reference Instrument***

**Anonymous Referee #2, 06 August 2023**

*We thank referee#2 for her/his complete and thoughtful review of our manuscript and providing thoughtful comments and suggestions that have helped us improve this manuscript. We also thank Editor Birgit Hassler for handling our paper and coordinating the reviews. Our responses to reviewer comments are provided below in red italic text.*

**RC2**: **'Comment on egusphere-2023-1466'**

**General comments**

The submitted manuscript reports, with great care and thoroughness, recent results that address two very long-standing issues in the ozonesonde community, namely the existence of a slow reaction component as well as a constant component to the traditional concept of background current, and also the separation of the traditional "pump correction factors" into a true pump efficiency and a second calibration function for different combinations of sondes and solutions.

While neither of these concepts are new the authors have performed an important job by their careful testing and now, with this manuscript, documentation, and I don't hesitate to recommend acceptance by AMT with only very minor revisions.

It is also very pleasing to see additional value being extracted from pervious JOSIE studies and I suspect there is yet more value still to be gained from analysis of the many experiments conducted at the WCCOS over the years.

The analysis has been very carefully performed and I could not find any examples of errors or statements that seemed illogical.

The manuscript is very long and at time, quite repetitive. Personally, I would have preferred a shorter study with a tighter focus, but I accept that there also some advantages to the very thorough approach taken by the authors here.

I think it is to the authors' credit that the results on the fast and slow responses are carefully compared with those of Voemel et al. 2020 and the differences clearly noted and discussed. This is very good scientific practice. (I see that "stimulating discussions" are noted in the acknowledgements).

Perhaps surprisingly, the manuscript also contains numerous mentions of effects that cannot yet be properly accounted for, or where only speculative causes can be postulated, eg lines 230, 235-236,551-559, 611-614, 656, 870-872, 894-895, 1157-1160, 1196-1198, 1220-1223, 1246-1248. It is good that the authors are candid about the lack of understanding of these topics but I hope that further work will help improve the situation.

I assume, however, that these remaining unknown effects can all be accounted for empirically by comparison to the photometer, even if the mechanism is not understood?

I have two very general questions I would like to pose to the authors.

Firstly, it seems to me that the authors consider the pump efficiencies as a function of pressure determined by Nakano and Morofuji 2023 to be, essentially, "correct". I would have

thought that these values might well have varied considerably over the very many years that ECC sondes have been manufactured and flown – could the authors please comment on that point?

>>> *For the Nakano and Morofuji (2023) pump efficiency table we used the average pump efficiencies and their uncertainties derived over all individual pump flow calibrations (1387 samples) which are listed in Table 2 of their publication and covers the entire period 2009 - 2022. Within 1-2% uncertainties theses average average values are rather stable over time, however, Nakano and Morofuji demonstrated that the pump efficiencies at pressures below 30 hPa can usually vary by a 1-2 percent between production badges. An exception is the period 2014-2018 whereby a systematic drop of 1-3 % in the pump efficiencies at reduced pressures below 30 hPa were observed. It is clear that in such cases when the actual pump efficiency values deviate significantly from the average Nakano and Morofuji pump efficiency then the calibration functions reported in Table 4 of this study have to be adjusted accordingly”. To make this clear we therefore included at Line 865 an extra sentence stating this.*

*“Of course, the numbers are for the calibration functions, i.e. the conversion efficiencies, directly linked to the pump efficiency values used, and it is assumed here that the average pump efficiency values from Nakano and Morofuji (2023) in Table 4 are correct within their uncertainties and representative of this study. However, if known pump efficiency values have changed over time, the calibration functions must be adjusted accordingly.”*

*A similar statement we have included in the conclusion in Chapter 8 at Line 1140*

Secondly, do you expect the results also to hold in the light of fluctuations in the manufacturing process of the ozonesondes, eg the "drop-off" reported by Stauffer et al. which affected ENSCI sondes and recent anecdotal reports in the community of poor production quality of SPC ozonesondes?

>>> *Of course applying the TRCC methodology neither the conventional method cannot avoid the TCO-drop or other inconveniences in the quality of the recorded ozonesonde data that have occurred in the past or in the future. However, in all these cases it had been shown again and again that to achieve the best possible quality and stability of the long term ozonesonde records it is of crucial importance the existence of a good functioning QA (Quality Assurance) management plan that covers all aspects of ozonesoundings in practice: (i) good functioning and stable instruments (manufacturing) (ii) good quality and well maintained ground equipment at the sounding station; (iii) unified SOPs (Standard Operating Procedures); (iv) and on top well trained and motivated personal who do the ozone soundings. Over the last 25 years large QA efforts have been done by the ozonesonde community within GAW-NDACC-SHADOZ and GRUAN, such as. (i) experimental activities (e.g. JOSIE 1996-2017, BESOS-2004); (ii) assessment to harmonize operating procedures (ASOPOS 1.0); (iii) homogenization of long term ozonesonde records (O3S-DQA); (iv) most recently the ASOPOS 2.0 (GAW Report No. 268). The ASOPOS 2.0 panel has also recognized that QA monitoring in quasi real time should be an important QA-component in the future. In this context a first example is the TCO drop study by Stauffer et al. (2020, 2022) by comparing sonde TCO data with satellite and ground based (Dobson/Brewer) TCO observations*

*But thanks for this question, for the sake of clarity we have included a short statement in the conclusions on this topic.*

Apart from these two questions, I only have very minor comments to make.

**Specific comments**

Lines 17-35 The abstract is almost all about the fast and slow responses, and it is not until line 31 that the new calibration functions are mentioned, even though they are also a very important result from this work. I suggest that the abstract be slightly re-worded so that the two main results appear earlier and more prominently.

*>>> We add an extra sentence in Line 24:*

*"The methodology resolves the slow and fast time responses of the ECC ozonesonde and in addition apply calibration functions to make the sonde data traceable to the JOSIE ozone reference UV-photometer (OPM).*

Line 47 Missing closing bracket

*>>> Done*

Line 52 I would prefer some references to the original literature here (Haagen Smit etc) rather than just a textbook. For the stratospheric ozone discussion which follows, many original references have been cited which I think is better.

*>>> We included the first Haagen-Smit, 1952 reference and changed the sentence by more focusing on the origin of the photochemical summer smog. Further, we think in the previous lines (43-51) the photochemistry of ozone in the troposphere has been already discussed in a balanced way comparing to the stratospheric ozone discussion part.*

Line 75 I would prefer a word such as "approximately" to the use of the tilde symbol here in the text.

*>>> Done*

Line 76 Would it be possible to include a map of the sixty stations? This would help the reader to understand that the work is relevant to a true global network. I appreciate it can be difficult to determine whether a station is "currently active".

*>>> We have considered that too, but finally decided not to include an extra figure because of the large number of figures already excisting. However, therefore we have explicitely referred to figure 1-2 of the new GAW-Report No. 268 which is shown such a map of all global ozonesounding stations that are operational in the global network.*

Line 77 Insert "the" before "stratosphere"

*>>> Done*

Line 132 Please re-word "giving artefact low readings". Do you mean "artificially low" or "unreliably low" or something like that?

*>>> Thanks for alerting. The sentence has been completely changed into: "A special challenge of tropical soundings is that near the tropopause the ozone concentrations can be*

*very low (Thompson et al., 2007b), such that the signal to noise is very small, causing large relative uncertainties in the ozonesonde readings (Smit et al. 2007)".*

Line 175 Insert "the" before "redox"

*>>> Done*

Line 189 Replace "convert" with "is converted to"

*>>> Done*

Line 209 I would prefer a sentence here explaining to the reader how this result was obtained without having to go back to the cited studies – ie how is it known that the absorption efficiency is 1.0 within 1% for all ECC ozonesondes?

*>>> We added an extra sentence in Line 209:*

*"This was confirmed by Davies et al. (2003), who determined experimentally at different pressures in a vacuum tank. the absorption efficiency ηA from the responses of two ECC-sondes connected in series."*

Lines 302-308 I found this a very helpful paragraph!

Lines 351-355 Figure 1 is a little bit confusing, because in part (a) the fast response is clearly extrapolated from the first three points, but in part (b) it's not obvious how the solid red line has been obtained and the reader might be given the impression that there is also a third, even slower response.

*>>> You are right. We have modified the figure by adding a new curve: IM(t)-IB0 and at t=10 min. we add the 25 min slow response (i.e. 25 min decay) that match very close the measured IM(t)-IB0 curve. We have changed the text in the manuscript accordingly.*

Line 435 Please re-word "approximately exponentially to a change in U"

*>>> Thanks to alert to re-word: correct is that the measured signal U is approximately proportional to its change over time dU/dt. We have correct in "approximately proportional to a change in time of U".*

Line 443 You say, in effect, that the method assumes that the time-step is small compared to the rate of change of ozone in the atmosphere, but you don't say whether this is a good assumption.

*>>> Sentence has been revised into: "In case the time step $\Delta t_k$ is chosen small relative to the response time $\tau$. then it can be assumed that the "true" (ambient) signal $U_a$ is quasi-stationary during time step $\Delta t_k$ such that $U_a(t_k) = U_a(t_{k-1})$"*

Line 609 Please re-word "for 50% contributing"

*>>> Done*

Line 655 Please re-word "may display be due to processes"

*>>> Done*

Line 656 "min" -> minutes

>>> *Done*

Line 803 I would prefer "factor of 2" and "factor of 3"

>>> *Done*

Line 805 Remove "been" – and I think replace with "significantly" or "non-negligibly"

>>> *Done*

Line 962 I would prefer "factor of 2"

>>> *Done*

Line 1048 Please reword – perhaps "In addition, the total uncertainty of the conventional method is also shown (dashed red lines)."

>>> *Done*

Line 1053 Please re-word "remote air conditions". (Or delete it altogether, I don't think you need to say it at all, and the sentence is too long.)

>>> *Done*

Line 1056 Please re-word "strongly restricted to its detection limit" – what does that really mean?

>>> *We replaced "restricted to" by "determined by"*

Lines 1060-1065 It's very good that you have shown how the procedure influences the results in these three different situations.

>>> *Done*

Line 1135 Please re-word "Essential thereby is .. "

>>> *Done*

Line 1136 I am not sure that you have smoothed it enough?

>>> *Done*

Line 1165 Delete the two "is"

>>> *Done*

Line 1188 "resp" should be "respective"

>>> *Done*